# Neuro-symbolic Learning Yielding Logical Constraints

**Zenan Li**[1]    **Yunpeng Huang**[1]    **Zhaoyu Li**[2]
**Yuan Yao**[1]    **Jingwei Xu**[1]    **Taolue Chen**[3]    **Xiaoxing Ma**[1]    **Jian Lü**[1]

[1]State Key Lab of Novel Software Technology, Nanjing University, China
[2]Department of Computer Science, University of Toronto, Canada
[3]School of Computing and Mathematical Sciences, Birkbeck, University of London, UK
`{lizn, hyp}@smail.nju.edu.cn, zhaoyu@cs.toronto.edu,`
`t.chen@bbk.ac.uk, {y.yao,jingweix,xxm,lj}@nju.edu.cn`

## Abstract

Neuro-symbolic systems combine neural perception and logical reasoning, representing one of the priorities of AI research. End-to-end learning of neuro-symbolic systems is highly desirable, but remains to be challenging. Resembling the distinction and cooperation between System 1 and System 2 of human thought (à la Kahneman), this paper proposes a framework that fuses neural network training, symbol grounding, and logical constraint synthesis to support learning in a weakly supervised setting. Technically, it is cast as a game with two optimization problems which correspond to neural network learning and symbolic constraint learning respectively. Such a formulation naturally embeds symbol grounding and enables the interaction between the neural and the symbolic part in both training and inference. The logical constraints are represented as cardinality constraints, and we use the trust region method to avoid degeneracy in learning. A distinguished feature of the optimization lies in the Boolean constraints for which we introduce a difference-of-convex programming approach. Both theoretical analysis and empirical evaluations substantiate the effectiveness of the proposed framework.

## 1   Introduction

Perception and reasoning serve as fundamental human abilities that are intrinsically linked within the realm of human intelligence [Kahneman, 2011, Booch et al., 2021]. The objective of our study is to develop a learning framework for neuro-symbolic systems (e.g., the one illustrated in Figure 1), enabling simultaneous learning of neural perception and symbolic reasoning.

The merit of neuro-symbolic learning lies in resembling the integration of System 1 and System 2 of human minds [Kahneman, 2011, Yoshua and Gary, 2020, LeCun, 2022]. First, it eliminates unnatural and sometimes costly human labeling of the latent variables and conducts learning in an end-to-end fashion. Second, it generates not only a neural network for perception, but also a set of explicit (symbolic) constraints enabling exact and interpretable logical reasoning. Last but not least, the mutually beneficial interaction between the neural and the symbolic parts during both training and inference stages potentially achieves better performance than separated learning approaches.

However, existing approaches do not provide an adequate solution to the problem. They either (1) are not end-to-end, i.e., human intervention is employed to label the latent $\mathbf{z}$ so that the task can be divided into a purely neural subtask of image classification and a purely symbolic subtask of constraint solving, or (2) are not interpretable, i.e., can only approximate symbolic reasoning with neural network but without explicit logical constraints generated (e.g., Wang et al. [2019], Yang et al. [2023]), which inevitably sacrifices the exactness and interpretability of symbolic reasoning, resulting in an inaccurate black-box predictor but not a genuine neuro-symbolic system.

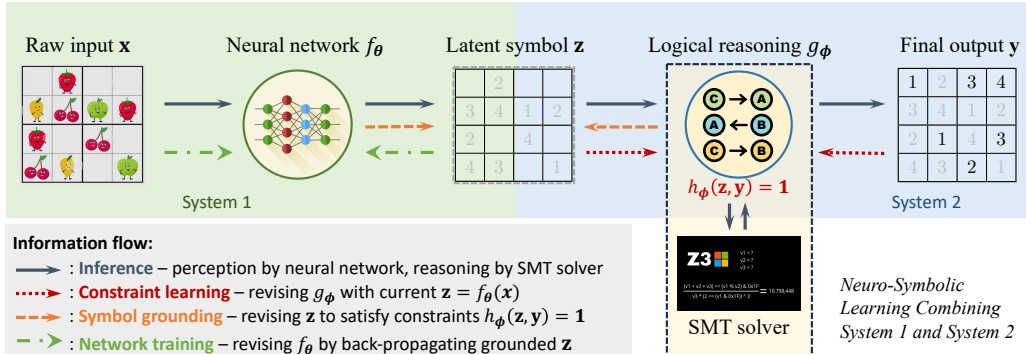

Figure 1: An example of neuro-symbolic learning for visual SudoKu solving. In this task, the neural network is employed to transform the puzzle image (strawberry etc.) into its corresponding symbols, while symbolic reasoning is utilized to produce the puzzle's solution. Importantly, the neuro-symbolic learning task is framed in a *weakly supervised* setting, where only the raw input (the puzzle image $\mathbf{x}$) and the final output (the puzzle solution $\mathbf{y}$, but without numbers in $\mathbf{z}$) is observed.

We argue that end-to-end and interpretable neuro-symbolic learning is extremely challenging due to the *semantic and representation gaps* between the neural part and the symbolic part. The semantic gap caused by the latency of the intermediate symbol (i.e., $\mathbf{z}$ in Figure 1) makes the neural network training lack effective supervision and the logic constraint synthesis lack definite inputs. The representation gap between the differentiable neural network and the discrete symbol logic makes it difficult to yield explicit symbolic constraints given the continuous neural network parameters. Despite existing proposals mitigating some of these obstacles such as visual symbol grounding [Topan et al., 2021, Li et al., 2023], softened logic loss [Kimmig et al., 2012, Xu et al., 2018], semidefinite relaxation [Wang et al., 2019], etc., none of them realize the full merit of neuro-symbolic learning.

In this paper, we propose a new neuro-symbolic learning framework directly meeting the challenges. It bridges the semantic gap with an efficient symbol grounding mechanism that models the cooperative learning of both the neural network and the logical constraint as a bilevel optimization problem. It bridges the representation gap by employing difference-of-convex (DC) programming as a relaxation technique for Boolean constraints in the optimization. DC programming ensures the convergence to explicit logical constraints, which enables exact symbolic reasoning with powerful off-the-shell tools such as SAT/SMT solvers [Eén and Sörensson, 2006, Bailleux and Boufkhad, 2003, Bailleux et al., 2006] during the inference stage. In addition, to address degeneracy in logical constraint learning, i.e., the tendency to learn only trivial logical constraints (e.g., resulting in simple rules insufficient to solve SudoKu), we introduce an additional trust region term [Boyd et al., 2004, Conn et al., 2000], and then employ the proximal point algorithm in the learning of logical constraints.

We provide a theoretical analysis of the convergence of our algorithm, as well as the efficacy of the DC relaxation in preserving the exactness and the trust region in preventing degeneracy. Empirical evaluations with four tasks, viz. Visual Sudoku Solving, Self-Driving Path Planning, Chained XOR, and Nonograms, demonstrate the new learning capability and the significant performance superiority of the proposed framework.

*Organization.* Section 2 formulates our neuro-symbolic learning framework. Section 3 details the algorithm and theoretical analysis. Section 4 presents empirical evaluations. Section 5 covers related work. Section 6 discusses the limitations. Section 7 concludes the paper.

## 2  Neuro-symbolic Learning Framework

In this paper, we focus on end-to-end neuro-symbolic systems comprising two components: (1) neural network $f_{\boldsymbol{\theta}}\colon \mathcal{X} \to \mathcal{Z}$, which transforms the raw input $\mathbf{x} \in \mathcal{X}$ into a latent state $\mathbf{z} \in \mathcal{Z}$; and (2) symbolic reasoning $g_{\boldsymbol{\phi}}\colon \mathcal{Z} \to \mathcal{Y}$, which deduces the final output $\mathbf{y} \in \mathcal{Y}$ from state $\mathbf{z} \in \mathcal{Z}$. Both components are built simultaneously, taking only the input $\mathbf{x}$ and output $\mathbf{y}$ as supervision. We assume that $\mathcal{Z}$ and $\mathcal{Y}$ are represented by finite sets, and thus we can encode $\mathbf{z}$ and $\mathbf{y}$ by binary vectors using

one-hot encoding. For ease of discussion, we directly define $\mathcal{Z}$ and $\mathcal{Y}$ to be spaces $\mathcal{B}^u$ and $\mathcal{B}^v$ of Boolean vectors, respectively, where $\mathcal{B} = \{0, 1\}$.

Unlike existing work (e.g., SATNet [Wang et al., 2019]) that simulates logical reasoning in an *implicit* and *approximate* way via a network layer, our key insight is to learn *explicit* logical constraints $h_\phi \colon \mathcal{B}^{u+v} \to \mathcal{B}$ on $(\mathbf{z}, \mathbf{y})$ in the training phrase, which allow to perform *exact* reasoning by off-the-shelf constraint solvers (e.g., SMT solvers) in the inference phrase. Fig. 1 illustrates our neuro-symbolic learning framework.

The latent state $\mathbf{z}$ enables the interaction between neural perception and logical reasoning. If $\mathbf{z}$ were observable, we could perform the learning of neural network and logical constraint by solving the following two separate optimization problems:

$$\boldsymbol{\theta} = \arg\min_{\boldsymbol{\theta}} \mathbb{E}_{(\mathbf{x},\mathbf{z})\sim\mathcal{D}_1}[\ell_1(f_{\boldsymbol{\theta}}(\mathbf{x}), \mathbf{z})], \qquad \boldsymbol{\phi} = \arg\min_{\boldsymbol{\phi}} \mathbb{E}_{(\mathbf{z},\mathbf{y})\sim\mathcal{D}_2}[\ell_2(h_{\boldsymbol{\phi}}(\mathbf{z}, \mathbf{y}), 1)],$$

where $\ell_1(f_{\boldsymbol{\theta}}(\mathbf{x}), \mathbf{z})$ refers to the error between network prediction $f_{\boldsymbol{\theta}}(\mathbf{x})$ and the actual symbol $\mathbf{z}$, and $\ell_2(h_{\boldsymbol{\phi}}(\mathbf{z}, \mathbf{y}), 1)$ refers to the (un)satisfaction degree of the learned logical constraints $h_{\boldsymbol{\phi}}(\mathbf{z}, \mathbf{y}) = 1$.

Nevertheless, with $\mathbf{z}$ being latent, these two problems become tightly coupled:

$$\boldsymbol{\theta} = \arg\min_{\boldsymbol{\theta}} \mathbb{E}_{(\mathbf{x},\mathbf{y})\sim\mathcal{D}}[\ell_1(f_{\boldsymbol{\theta}}(\mathbf{x}), \mathbf{z})], \quad \boldsymbol{\phi} = \arg\min_{\boldsymbol{\phi}} \mathbb{E}_{(\mathbf{x},\mathbf{y})\sim\mathcal{D}}[\ell_2(h_{\boldsymbol{\phi}}(\mathbf{z}, \mathbf{y}), 1)],$$
$$\text{s.t.} \quad h_{\boldsymbol{\phi}}(\mathbf{z}, \mathbf{y}) = 1, \mathbf{z} \in \mathcal{Z}; \qquad\qquad \text{s.t.} \quad \mathbf{z} = f_{\boldsymbol{\theta}}(\mathbf{x}), \mathbf{z} \in \mathcal{Z}.$$

We further surrogate the constraint satisfaction by loss functions, and obtain essentially a game formulation as follows:

$$\boldsymbol{\theta} = \arg\min_{\boldsymbol{\theta}} \mathbb{E}_{(\mathbf{x},\mathbf{y})\sim\mathcal{D}}[\ell_1(f_{\boldsymbol{\theta}}(\mathbf{x}), \bar{\mathbf{z}})], \qquad\qquad \boldsymbol{\phi} = \arg\min_{\boldsymbol{\phi}} \mathbb{E}_{(\mathbf{x},\mathbf{y})\sim\mathcal{D}}[\ell_2(h_{\boldsymbol{\phi}}(\bar{\mathbf{z}}, \mathbf{y}), 1)],$$
$$\text{s.t.} \quad \bar{\mathbf{z}} = \arg\min_{\mathbf{z}\in\mathcal{Z}} \ell_2(h_{\boldsymbol{\phi}}(\mathbf{z}, \mathbf{y}), 1); \qquad \text{s.t.} \quad \bar{\mathbf{z}} = \arg\min_{\mathbf{z}\in\mathcal{Z}} \ell_1(f_{\boldsymbol{\theta}}(\mathbf{x}), \mathbf{z}). \tag{1}$$

Three players are involved in this game: neural network $f_{\boldsymbol{\theta}}$ and logical constraint $h_{\boldsymbol{\phi}}$ pursue optimal (prediction/satisfaction) accuracy, while $\mathbf{z}$ strives for the grounding that integrates the network prediction and logical reasoning.

## 2.1  Efficient and Effective Logical Constraint Learning

For *efficient* learning of logical constraints, we adopt cardinality constraint [Syrjänen, 2009, 2004, Fiorini et al., 2021] to represent logical constraint.[1] Cardinality constraints can be easily arithmetized, enabling the conventional optimization method and avoiding the computationally expensive model counting or sampling [Manhaeve et al., 2018, Xu et al., 2018, Li et al., 2020, 2023], which significantly boosts the learning efficiency. In addition, the conjunctive normal form and cardinality constraints can be easily converted from each other, ensuring not only the expressiveness of the learned constraints, but also the seamless compatibility with existing reasoning engines.

Formally, we denote the column concatenation of $\mathbf{z}$ and $\mathbf{y}$ by $(\mathbf{z}; \mathbf{y})$, and define a cardinality constraint $h_{\boldsymbol{\phi}}(\mathbf{z}, \mathbf{y}) := \boldsymbol{w}^\mathsf{T}(\mathbf{z}; \mathbf{y}) \in [b_{\min}, b_{\max}]$, where $\boldsymbol{\phi} = (\boldsymbol{w}, b_{\min}, b_{\max})$, $\boldsymbol{w} \in \mathcal{B}^{u+v}$ is a Boolean vector, and $b_{\min}, b_{\max} \in \mathcal{N}_+$ are two positive integers. Moreover, we can directly extend $h_{\boldsymbol{\phi}}$ to the matrix form representing $m$ logical constraints, i.e.,

$$h_{\boldsymbol{\phi}}(\mathbf{z}, \mathbf{y}) := \boldsymbol{W}(\mathbf{z}; \mathbf{y}) = (\boldsymbol{w}_1^\mathsf{T}(\mathbf{z}; \mathbf{y}), \ldots, \boldsymbol{w}_m^\mathsf{T}(\mathbf{z}; \mathbf{y})) \in [\boldsymbol{b}_{\min}, \boldsymbol{b}_{\max}],$$

where $\boldsymbol{\phi} = (\boldsymbol{W}, \boldsymbol{b}_{\min}, \boldsymbol{b}_{\max})$, $\boldsymbol{W} := (\boldsymbol{w}_1^\mathsf{T}; \ldots; \boldsymbol{w}_m^\mathsf{T}) \in \mathcal{B}^{m\times(u+v)}$, and $\boldsymbol{b}_{\min}, \boldsymbol{b}_{\max} \in \mathcal{N}_+^m$.

For *effective* learning of logical constraints, one must tackle the frequent problem of *degeneracy* that causes incomplete or trivial constraints[2]. First, for the bounding box $[\boldsymbol{b}_{\min}, \boldsymbol{b}_{\max}]$, any of its superset is a feasible result of logical constraint learning, but we actually expect the tightest one. To overcome this difficulty, we propose to use the classic mean squared loss, i.e., defining $\ell_2(h_{\boldsymbol{\phi}}(\mathbf{z}, \mathbf{y}), 1) = \|\boldsymbol{W}(\mathbf{z}; \mathbf{y}) - \boldsymbol{b}\|^2$, where $\boldsymbol{b}$ can be computed in optimization or pre-defined. Then, our logical constraint learning problem can be formulated as a Boolean least squares problem [Vandenberghe and Boyd, 2004]. The unbiased property of least squares indicates that $\boldsymbol{b} \approx (\boldsymbol{b}_{\min} + \boldsymbol{b}_{\max})/2$, and

---

[1] Cardinality constraints express propositional logic formulae by constraining the number of variables being true. For example, $x \vee y \vee \neg z$ is encoded as $x + y + \bar{z} \geq 1$.

[2] For example, suppose the targeted logical constraints are $\mathbf{z}_1 + \mathbf{z}_2 + \mathbf{y}_1 \geq 2$ and $\mathbf{z}_1 \geq 1$. Degeneracy happens if one only obtains a single constraint $\mathbf{z}_1 + \mathbf{z}_2 + \mathbf{y}_1 \geq 2$ or a trivial constraint $\mathbf{z}_1 \geq 0$.

the minimum variance property of least squares ensures that we can achieve a tight bounding box $[\boldsymbol{b}_{\min}, \boldsymbol{b}_{\max}]$ [Henderson, 1975, Björck, 1990, Hastie et al., 2009].

Second, it is highly possible that some of the $m$ distinct logical constraints as indicated by matrix $\boldsymbol{W}$ eventually degenerate to the same one during training, because the Boolean constraints and stochastic gradient descent often introduce some implicit bias [Gunasekar et al., 2017, Smith et al., 2021, Ali et al., 2020]. To mitigate this problem, we adopt the trust region method [Boyd et al., 2004, Conn et al., 2000], i.e., adding constraints $\|\boldsymbol{w}_i - \boldsymbol{w}_i^{(0)}\| \leq \lambda, i = 1, \ldots, m$, where $\boldsymbol{w}_i^{(0)}$ is a pre-defined centre point of the trust region. The trust region method enforces each $\boldsymbol{w}_i$ to search in their own local region. We give an illustrative figure in Appendix D to further explain the trust region method.

To summarize, using the penalty instead of the trust region constraint, we can formulate the optimization problem of logical constraint learning in (1) as

$$
\begin{aligned}
\min_{(\boldsymbol{W}, \boldsymbol{b})} \quad & \mathbb{E}_{(\mathbf{x}, \mathbf{y}) \sim \mathcal{D}}[\|\boldsymbol{W}(\bar{\mathbf{z}}; \mathbf{y}) - \boldsymbol{b}\|^2] + \lambda \|\boldsymbol{W} - \boldsymbol{W}^{(0)}\|^2, \\
\text{s.t.} \quad & \bar{\mathbf{z}} = \arg\min_{\mathbf{z} \in \mathcal{Z}} \ell_1(f_{\boldsymbol{\theta}}(\mathbf{x}), \mathbf{z}), \quad \boldsymbol{W} \in \mathcal{B}^{m \times (u+v)}, \quad \boldsymbol{b} \in \mathcal{N}_+^m.
\end{aligned}
\tag{2}
$$

## 2.2 Neural Network Learning in Tandem with Constraint Learning

A key challenge underlines end-to-end neuro-symbolic learning is *symbol grounding*, which is to tackle the chicken-and-egg situation between network training and logical constraint learning: training the network requires the supervision of symbol $\mathbf{z}$ that comes from solving the learned logical constraints, but the constraint learning needs $\mathbf{z}$ as input recognized by the trained network. Specifically, since matrix $\boldsymbol{W}$ is often underdetermined in high-dimensional cases, the constraint $\bar{\mathbf{z}} = \arg\min_{\mathbf{z} \in \mathcal{Z}} \ell_2(h_{\boldsymbol{\phi}}(\mathbf{z}, \mathbf{y}), 1) := \|\boldsymbol{W}(\mathbf{z}; \mathbf{y}) - \boldsymbol{b}\|^2$ often has multiple minimizers (i.e., multiple feasible groundings of $\mathbf{z}$), all of which satisfy the logical constraints $h_{\boldsymbol{\phi}}(\mathbf{z}, \mathbf{y}) = 1$. Moreover, matrix $\boldsymbol{W}$ is also a to-be-trained parameter, meaning that it is highly risky to determine the symbol grounding solely on the logical constraints.

To address these issues, instead of (approximately) enumerating all the feasible solutions via model counting or sampling [Manhaeve et al., 2018, Xu et al., 2018, Li et al., 2020, van Krieken et al., 2022, Li et al., 2023], we directly combine network prediction and logical constraint satisfaction to establish symbol grounding, owing to the flexibility provided by the cardinality constraints. Specifically, for given $\alpha \in [0, +\infty)$, the constraint in network learning can be rewritten as follows,

$$
\bar{\mathbf{z}} = \arg\min_{\mathbf{z} \in \mathcal{Z}} \|\boldsymbol{W}(\mathbf{z}; \mathbf{y}) - \boldsymbol{b}\|^2 + \alpha \|\mathbf{z} - f_{\boldsymbol{\theta}}(\mathbf{x})\|^2.
$$

The coefficient $\alpha$ can be interpreted as the preference of symbolic grounding for network predictions or logical constraints. For $\alpha \to 0$, the symbol grounding process can be interpreted as distinguishing the final symbol $\mathbf{z}$ from all feasible solutions based on network predictions. For $\alpha \to +\infty$, the symbol grounding process can be viewed as a "correction" step, where we revise the symbol grounding from network's prediction towards logical constraints. Furthermore, as we will show in Theorem 1 later, both symbol grounding strategies can finally converge to the expected results.

The optimization problem of network training in (1) can be written as

$$
\begin{aligned}
\min_{\boldsymbol{\theta}} \quad & \mathbb{E}_{(\mathbf{x}, \mathbf{y}) \sim \mathcal{D}}[\|\bar{\mathbf{z}} - f_{\boldsymbol{\theta}}(\mathbf{x})\|^2], \\
\text{s.t.} \quad & \bar{\mathbf{z}} = \arg\min_{\bar{\mathbf{z}} \in \mathcal{Z}} \|\boldsymbol{W}(\bar{\mathbf{z}}; \mathbf{y}) - \mathbf{b}\|^2 + \alpha \|\bar{\mathbf{z}} - f_{\boldsymbol{\theta}}(\mathbf{x})\|^2.
\end{aligned}
\tag{3}
$$

where we also use the mean squared loss, i.e., $\ell_1(f_{\boldsymbol{\theta}}(\mathbf{x}), \mathbf{z}) = \|f_{\boldsymbol{\theta}}(\mathbf{x}) - \mathbf{z}\|^2$, for compatibility.

## 3 Algorithms and Analysis

Our general framework is given by (1), instantiated by (2) and (3). Both optimizations contain Boolean constraints of the form $\|\boldsymbol{Q}\boldsymbol{u} - \boldsymbol{q}_1\|^2 + \tau \|\boldsymbol{u} - \boldsymbol{q}_2\|^2$ where $\boldsymbol{u}$ are Boolean variables.[3] We propose to relax these Boolean constraints by *difference of convex* (DC) programming [Tao and Hoai An, 1997, Yuille and Rangarajan, 2003, Lipp and Boyd, 2016, Hoai An and Tao, 2018]. Specifically, a

---

[3]In (2), for each logic constraint, $\boldsymbol{Q} = (\mathbf{z}; \mathbf{y})^{\mathsf{T}}, \boldsymbol{q}_1 = \boldsymbol{b}_i, \boldsymbol{q}_2 = \boldsymbol{w}_i^{(0)}$, and $\tau = \lambda$; in (3), $\boldsymbol{Q} = \boldsymbol{W}, \boldsymbol{q}_1 = \boldsymbol{b}, \boldsymbol{q}_2 = (f_{\boldsymbol{\theta}}(\mathbf{x}); \mathbf{y})$, and $\tau = \alpha$.

Boolean constraint $u \in \{0,1\}$ can be rewritten into two constraints of $u - u^2 \geq 0$ and $u - u^2 \leq 0$. The first constraint is essentially a box constraint, i.e., $u \in [0,1]$, which is kept in the optimization. The second one is concave, and we can *equivalently* add it as a penalty term, as indicated by the following proposition [Bertsekas, 2015, Hansen et al., 1993, Le Thi and Ding Tao, 2001] .

**Proposition 1.** *Let $\boldsymbol{e}$ denote the all-one vector. There exists $t_0 \geq 0$ such that for every $t > t_0$, the following two problems are equivalent, i.e., they have the same optimum.*

$$(P) \quad \min_{\boldsymbol{u} \in \{0,1\}^n} q(\boldsymbol{u}) := \|\boldsymbol{Q}\boldsymbol{u} - \boldsymbol{q}_1\|^2 + \tau\|\boldsymbol{u} - \boldsymbol{q}_2\|^2,$$

$$(P_t) \quad \min_{\boldsymbol{u} \in [0,1]^n} q^t(\boldsymbol{u}) := \|\boldsymbol{Q}\boldsymbol{u} - \boldsymbol{q}_1\|^2 + \tau\|\boldsymbol{u} - \boldsymbol{q}_2\|^2 + t(\boldsymbol{e}^\mathsf{T}\boldsymbol{u} - \boldsymbol{u}^\mathsf{T}\boldsymbol{u}).$$

*Remarks.* We provide more details, including the setting of $t_0$, in Appendix A.

However, adding this penalty term causes non-convexity. Thus, DC programming further linearizes the penalty $u - u^2 \approx \tilde{u} - \tilde{u}^2 + (u - \tilde{u})(1 - 2\tilde{u})$ at the given point $\tilde{u}$, and formulates the problem in Proposition 1 as

$$\min_{\boldsymbol{u} \in [0,1]^n} \|\boldsymbol{Q}\boldsymbol{u} - \boldsymbol{q}_1\|^2 + \tau\|\boldsymbol{u} - \boldsymbol{q}_2\|^2 + t(\boldsymbol{e} - 2\tilde{\boldsymbol{u}})^\mathsf{T}\boldsymbol{u}.$$

By applying this linearization, we achieve a successive convex approximation to the Boolean constraint [Razaviyayn, 2014], ensuring that the training is more stable and globally convergent [Lipp and Boyd, 2016]. Furthermore, instead of fixing the coefficient $t$, we propose to gradually increase it until the Boolean constraint is fully satisfied, forming an "annealing" procedure. We illustrate the necessity of this strategy in the following proposition [Beck and Teboulle, 2000, Xia, 2009].

**Proposition 2.** *A solution $\boldsymbol{u} \in \{0,1\}^n$ is a stationary point of $(P_t)$ if and only if*

$$[\nabla q(\boldsymbol{u})]_i(1 - 2\boldsymbol{u}_i) + t \geq 0, \quad i = 1, \ldots, n.$$

*Then, if $\boldsymbol{u} \in \{0,1\}^n$ is a global optimum of $(P)$ (as well as $(P_t)$), it holds that*

$$[\nabla q(\boldsymbol{u})]_i(1 - 2\boldsymbol{u}_i) + \rho_i \geq 0, \quad i = 1, \ldots, n,$$

*where $\rho_i$ is the $i$-th diagonal element of $(\boldsymbol{Q}^\mathsf{T}\boldsymbol{Q} + \tau\boldsymbol{I})$.*

*Remarks.* The proposition reveals a trade-off of $t$: a larger $t$ encourages exploration of more stationary points satisfying the Boolean constraints, but a too large $t$ may cause the converged point to deviate from the optimality of (P). Therefore, a gradual increase of $t$ is sensible for obtaining the desired solution. Moreover, the initial minimization under small $t$ results in a small gradient value (i.e., $|[\nabla q(\boldsymbol{u})]_i|$), thus a Boolean stationary point can be quickly achieved with a few steps of increasing $t$.

## 3.1 Algorithms

For a given dataset $\{(\mathbf{x}_i, \mathbf{y}_i)\}_{i=1}^N$, $\mathbf{X} = (\mathbf{x}_1, \ldots, \mathbf{x}_N)$ and $\mathbf{Y} = (\mathbf{y}_1, \ldots, \mathbf{y}_N)$ represent the data matrix and label matrix respectively, and $f_{\boldsymbol{\theta}^{(k)}}(\mathbf{X})$ denotes the network prediction at the $k$-th iteration.

**Logical constraint learning.** Eliminating the constraint in (2) by letting $\mathbf{Z} = f_{\boldsymbol{\theta}^{(k)}}(\mathbf{X})$, the empirical version of the logical constraint learning problem at the $(k+1)$-th iteration is

$$\min_{(\boldsymbol{W}, \boldsymbol{b})} \sum_{i=1}^m \|(f_{\boldsymbol{\theta}^{(k)}}(\mathbf{X}); \mathbf{Y})\boldsymbol{w}_i - \boldsymbol{b}_i\|^2 + \lambda\|\boldsymbol{w}_i - \boldsymbol{w}_i^{(0)}\|^2 + t_1(\boldsymbol{e} - 2\boldsymbol{w}_i^{(k)})^\mathsf{T}\boldsymbol{w}_i,$$

where $\boldsymbol{w}_1^{(k)}, \ldots, \boldsymbol{w}_m^{(k)}$ are parameters of logical constraints at the $k$-th iteration. In this objective function, the first term is the training loss of logical constraint learning, the second term is the trust region penalty to avoid degeneracy, and the last term is the DC penalty of the Boolean constraint.

To solve this problem, we adopt the proximal point algorithm (PPA) [Rockafellar, 1976, Parikh et al., 2014, Rockafellar, 2021], as it overcomes two challenges posed by stochastic gradient descent. First, stochastic gradient descent has an implicit inductive bias [Gunasekar et al., 2017, Ali et al., 2020, Zhang et al., 2021, Smith et al., 2021], causing different $\boldsymbol{w}_i, i = 1, \ldots, m$, to converge to a singleton. Second, the data matrix $(f_{\boldsymbol{\theta}^{(k)}}(\mathbf{X}); \mathbf{Y})$ is a 0-1 matrix and often ill-conditioned, leading to diverse or slow convergence rates of stochastic gradient descent.

**Algorithm 1** Neuro-symbolic Learning Procedure

---

Set step sizes $(\gamma, \eta)$, and penalty coefficients $(\lambda, t_1, t_2)$.
Randomly generate an initial matrix $\boldsymbol{W}^{(0)}$ under $[0, 1]$ uniform distribution.
**for** $k = 0, 1, \ldots, K$ **do**
    Randomly draw a batch $\{(\mathbf{x}_i, \mathbf{y}_i)\}_{i=1}^{N}$ from training data.
    Compute the predicted symbol $\mathbf{z}_i = f_{\boldsymbol{\theta}^{(k)}}(\mathbf{x}_i), i = 1, \ldots, N.$      ▷ *Network prediction*
    Update $(\boldsymbol{W}, \boldsymbol{b})$ from $(\mathbf{z}_i, \mathbf{y}_i), i = 1, \ldots, m$, by PPA update (Eq. 4).     ▷ *Constraint learning*
    Correct the symbol grounding $\bar{\mathbf{z}}$ from $\mathbf{z}_i$ to logical constraints $h_{\phi}$ (Eq. 5). ▷ *Symbol grounding*
    Update $\boldsymbol{\theta}$ from $(\mathbf{x}_i, \bar{\mathbf{z}}_i), i = 1, \ldots, m$, by SGD update (Eq. 6).      ▷ *Network training*
    **if** $\mathbf{W} \notin \mathcal{B}^{m \times (u+v)}/\bar{\mathbf{Z}} \notin \mathcal{B}^{N \times (u+v)}$ **then**
        Increasing $t_1/t_2$.                               ▷ *Enforcing DC penalty*
    **end if**
**end for**
Estimate $(\boldsymbol{b}_{\min}, \boldsymbol{b}_{\max})$ based on network $f_{\boldsymbol{\theta}}$ and logical constraints $h_{\phi}$ from training data.

---

Given $(\boldsymbol{W}^{(k)}, \boldsymbol{b}^{(k)})$ at the $k$-th iteration, the update of PPA can be computed by

$$\boldsymbol{W}^{(k+1)} = (\boldsymbol{M} + (\lambda + \frac{1}{\gamma})\boldsymbol{I})^{-1}\Big((f_{\boldsymbol{\theta}^{(k)}}(\mathbf{X}); \mathbf{Y})^{\mathsf{T}}\boldsymbol{b}^{(k)} + \lambda\boldsymbol{W}^{(0)} + (\frac{1}{\gamma} + t_1)\boldsymbol{W}^{(k-1)} - \frac{t_1}{2}\boldsymbol{E}\Big),$$

$$\boldsymbol{b}^{(k+1)} = (1 + \frac{1}{\gamma})^{-1}(\boldsymbol{W}^{(k)}(f_{\boldsymbol{\theta}^{(k)}}(\mathbf{X}); \mathbf{Y})^{\mathsf{T}}\boldsymbol{e} + \frac{1}{\gamma}\boldsymbol{b}^{(k)}), \tag{4}$$

where $\gamma > 0$ is the step size of PPA, $\boldsymbol{E}$ is an all-ones matrix, and $\boldsymbol{M} = (f_{\boldsymbol{\theta}^{(k)}}(\mathbf{X}); \mathbf{Y})^{\mathsf{T}}(f_{\boldsymbol{\theta}^{(k)}}(\mathbf{X}); \mathbf{Y})$. Note that the computation of matrix inverse is required, but it is not an issue because $\boldsymbol{M}$ is positive semidefinite, and so is the involved matrix, which allows the use of Cholesky decomposition to compute the inverse. Moreover, exploiting the low rank and the sparsity of $\boldsymbol{M}$ can significantly enhance the efficiency of the computation.

**Neural network training.** By adding the DC penalty, the constraint (i.e., symbol grounding) in (3) is

$$\bar{\mathbf{z}} = \arg\min \|\boldsymbol{W}(\bar{\mathbf{z}}; \mathbf{y}) - \boldsymbol{b}\|^2 + \alpha\|\bar{\mathbf{z}} - f_{\boldsymbol{\theta}^{(k)}}(\mathbf{x})\|^2 + t_2(\boldsymbol{e} - 2(\bar{\mathbf{z}}^{(k)})^{\mathsf{T}})\bar{\mathbf{z}}.$$

We can also compute the closed-form solution, i.e.,

$$\bar{\mathbf{Z}} = \left(\boldsymbol{b}\boldsymbol{W} + (\alpha + t_2)\bar{\mathbf{Z}}^{(k)} + \alpha f_{\boldsymbol{\theta}^{(k)}}(\mathbf{X}) - \frac{t_2}{2}\boldsymbol{E}\right)\left(\boldsymbol{W}^{\mathsf{T}}\boldsymbol{W} + \alpha\boldsymbol{I}\right)^{-1}. \tag{5}$$

Similarly, the low-rank and sparsity properties of $\boldsymbol{W}^{\mathsf{T}}\boldsymbol{W}$ ensure an efficient computation of matrix inverse. Finally, the parameter $\boldsymbol{\theta}$ of network is updated by stochastic gradient descent,

$$\boldsymbol{\theta}^{(k+1)} = \boldsymbol{\theta}^{(k)} - \eta\nabla_{\boldsymbol{\theta}}f_{\boldsymbol{\theta}^{(k)}}(\mathbf{x}_i)\sum_{i=1}^{N}(\bar{\mathbf{z}}_i - f_{\boldsymbol{\theta}^{(k)}}(\mathbf{x}_i)). \tag{6}$$

The overall algorithm is summarized in Algorithm 1, which mainly involves three iterative steps: (1) update the logical constraints by combining the network prediction and observed output; (2) correct the symbol grounding by revising the prediction to satisfy logical constraints; (3) update the neural network by back-propagating the corrected symbol grounding with observed input.

### 3.2 Theoretical Analysis

**Theorem 1.** *With an increasing (or decreasing) $\alpha$, the constraint learning and network training performed by Algorithm 1 converge to the stationary point of (2) and (3), respectively. Specifically, it satisfies*

$$\mathbb{E}[\|\nabla_{\boldsymbol{\theta}}\ell_1^k(\boldsymbol{\theta}^k)\|^2] = \mathcal{O}(\frac{1}{\sqrt{K+1}}), \quad and \quad \mathbb{E}[\|\nabla_{\boldsymbol{\phi}}\ell_2(\boldsymbol{\phi}^k)\|^2] = \mathcal{O}(\frac{1}{\sqrt{K+1}}).$$

*Remarks.* The proof and additional results can be found in Appendix B. In summary, Theorem 1 confirms the convergence of our neuro-symbolic learning framework and illustrates its theoretical

complexity. Furthermore, note that an increase (or decrease) in $\alpha$ indicates a preference for correcting the symbol grounding over network prediction (or logical constraint learning). In practice, we can directly set a small (or large) enough $\alpha$ instead.

Next, we analyze the setting of centre points $\boldsymbol{W}^{(0)}$ in the trust region penalty as follows.

**Theorem 2.** *Let $\boldsymbol{w}_1^{(0)} \in [0,1]^n$ and $\boldsymbol{w}_2^{(0)} \in [0,1]^n$ be two initial points sampled from the uniform distribution. For given $t \geq 0$, the probability that the corresponding logical constraints $\phi_1$ and $\phi_2$ converge to the same (binary) stationary point $\phi$ satisfies*

$$\Pr(\boldsymbol{\phi}_1 = \boldsymbol{\phi}, \boldsymbol{\phi}_2 = \boldsymbol{\phi}) \leq \prod_{i=1}^{n} \min \left\{ \frac{1}{2\lambda}([\nabla q(\boldsymbol{u})]_i(1 - 2\boldsymbol{u}_i) + t), 1 \right\}^2.$$

*Remarks.* The proof is in Appendix C. In a nutshell, Theorem 2 shows that the probability of $\boldsymbol{w}_1$ and $\boldsymbol{w}_2$ degenerating to the same logical constraint can be very small provided suitably chosen $\lambda$ and $t$. Note that $\lambda$ and $t$ play different roles. As shown by Proposition 2, the coefficient $t$ in DC penalty ensures the logical constraint learning can successfully converge to a sensible result. The coefficient $\lambda$ in trust region penalty enlarges the divergence of convergence conditions between two distinct logical constraints, thereby preventing the degeneracy effectively.

## 4 Experiments

We carry out experiments on four tasks, viz., chained XOR, Nonogram, visual Sudoku solving, and self-driving path planning. We use Z3 SMT (MaxSAT) solver [Moura and Bjørner, 2008] for symbolic reasoning. Other implementation details can be found in Appendix E. The experimental results of chained XOR and Nonogram tasks are detailed in Appendix F due to the space limit. The code is available at `https://github.com/Lizn-zn/Nesy-Programming`.

### 4.1 Visual Sudoku Solving

**Datasets.** We consider two $9 \times 9$ visual SudoKu solving datasets, i.e., the SATNet dataset [Wang et al., 2019, Topan et al., 2021][4] and the RRN dataset [Yang et al., 2023], where the latter is more challenging (17 - 34 versus 31 - 42 given digits in each puzzle). Both datasets contain 9K/1K training/test examples, and their images are all sampled from the MNIST dataset. We typically involve two additional transfer tasks, i.e., training the neuro-symbolic system on SATNet dataset (resp. RRN dataset), and then evaluating the system on RRN dataset (resp. SATNet dataset).

**Baselines.** We compare our method with four state-of-the-art methods, i.e., RRN [Palm et al., 2018], SATNet [Wang et al., 2019], SATNet* [Topan et al., 2021], and L1R32H4 [Yang et al., 2023]. RRN is modified to match visual SudoKu as done by Yang et al. [2023]. SATNet* is an improved version of SATNet that addresses the symbol grounding problem by introducing an additional pre-clustering step. As part of our ablation study, we introduce two variants of our method (NTR and NDC) where NTR removes the trust region penalty (i.e., setting $\lambda = 0$), and NDC removes the DC penalty (i.e., fixing $t_1 = t_2 = 0$) and directly binarizes $(\boldsymbol{W}, \boldsymbol{b})$ as the finally learned logical constraints.

**Results.** We report the accuracy results (i.e., the percentage of correctly recognized boards, correctly solved boards, and both) in Table 1.[5] A more detailed version of our experimental results is given in Appendix F. The results show that our method significantly outperforms the existing methods in all cases, and both trust region penalty and DC penalty are critical design choices. The solving accuracy is slightly higher than the perception accuracy, as the MaxSAT solver may still solve the problem correctly even when the perception result is wrong. Notably, our method precisely learns all logical constraints,[6] resulting in a logical reasoning component that (1) achieves full accuracy when the neural perception is correct; (2) ensures robust results on transfer tasks, in comparison to the highly sensitive existing methods.

---

[4]The SATNet dataset is originally created by Wang et al. [2019]. However, Topan et al. [2021] point out that the original dataset has the label leakage problem, which was fixed by removing the labels of given digits.

[5]We successfully reproduce the baseline methods, achieving consistent results with Yang et al. [2023].

[6]We need 324 ($= 9 \times 9 \times 4$: each row/column/block should fill 1 - 9, and each cell should be in 1 - 9) ground-truth cardinality constraints for the Sudoku task, whose rank is 249. After removing redundant constraints, our method learns exact 324 logical constraints that are one-to-one corresponding to the ground-truth.

Table 1: Accuracy results (%) of visual Sudoku task. Our method performs the best.

| Method | SATNet dataset | | | RRN dataset | | | SATNet → RRN | | |
|---|---|---|---|---|---|---|---|---|---|
| | Percep. | Solving | Total | Percep. | Solving | Total | Percep. | Solving | Total |
| RRN | 0.0 | 0.0 | 0.0 | 0.0 | 0.0 | 0.0 | 0.0 | 0.0 | 0.0 |
| SATNet | 0.0 | 0.0 | 0.0 | 0.0 | 0.0 | 0.0 | 0.0 | 0.0 | 0.0 |
| SATNet* | 72.7 | 75.9 | 67.3 | 75.7 | 0.1 | 0.1 | 80.8 | 1.4 | 1.4 |
| L1R32H4 | 94.1 | 91.0 | 90.5 | 87.7 | 65.8 | 65.7 | 84.8 | 21.3 | 21.3 |
| NTR | 87.4 | 0.0 | 0.0 | 91.4 | 3.9 | 3.9 | 90.2 | 0.0 | 0.0 |
| NDC | 79.9 | 0.0 | 0.0 | 88.0 | 0.0 | 0.0 | 86.1 | 0.0 | 0.0 |
| Ours | **95.5** | **95.9** | **95.5** | **93.1** | **94.4** | **93.1** | **93.9** | **95.2** | **93.9** |

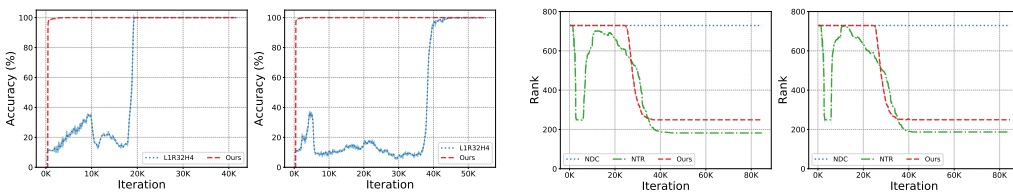

Figure 2: Training curves of accuracy (left) and rank (right). Our method significantly boosts the efficiency of symbol grounding, and accurately converges to ground-truth constraints.

Furthermore, we plot some training curves in Figure 2. The left two figures depict the training curves of neural perception accuracy on the SATNet dataset and RRN dataset, which demonstrate the extremely higher efficiency of our method in symbol grounding compared with the best competitor L1R32H4. We also compute the rank of matrix $W$ to evaluate the degeneracy of logical constraint learning. The results are presented in the right two figures, illustrating that logical constraints learned by our method are complete and precise. In contrast, the ablation methods either fail to converge to the correct logical constraints or result in a degenerate outcome.

### 4.2 Self-driving Path Planning

**Motivation.** Self-driving systems are fundamentally neuro-symbolic, where the primary functions are delineated into two components: object detection empowered by neural perception and path planning driven by symbolic reasoning. Neuro-symbolic learning has great potential in self-driving, e.g., for learning from demonstrations [Schaal, 1996] and to foster more human-friendly driving patterns [Sun et al., 2021, Huang et al., 2021].

**Datasets.** We simulate the self-driving path planning task based on two datasets, i.e., Kitti [Geiger et al., 2013] and nuScenes [Caesar et al., 2020]. Rather than provide the label of object detection, we only use planning paths as supervision. To compute planning paths, we construct obstacle maps with $10 \times 10$ grids, and apply the $A^*$ algorithm with fixed start points and random end points. Note that Kitti and nuScenes contain 6160/500 and 7063/600 training/test examples, respectively, where nuScenes is more difficult (7.4 versus 4.6 obstacles per image on average).

**Baselines.** We include the best competitor L1R32H4 [Yang et al., 2023] in the previous experiment as comparison. Alongside this, we also build an end-to-end ResNet model (denoted by ResNet) [He et al., 2016] and an end-to-end recurrent transformer model (denoted by RTNet) [Hao et al., 2019]. These models take the scene image, as well as the start point and the end point, as the input, and directly output the predicted path. Finally, as a reference, we train a ResNet model with direct supervision (denoted by SUP) by using labels of object detection, and the logical reasoning is also done by the $A^*$ algorithm.

**Results.** We include the $F_1$ score of predicted path grids, the collision rate of the planning path, and the distance error between the shortest path and the planning path (only computed for safe paths) in Table 2. The results show that our method achieves the best performance on both datasets, compared with the alternatives. Particularly, the existing state-of-the-art method L1R32H4 fails on this task,

Table 2: Results of self-driving path planning task. Our method performs the best.

| Method | Kitti dataset | | | nuScenes | | |
|---|---|---|---|---|---|---|
| | $F_1$ score ↑ | Coll. rate ↓ | Dist. Err. ↓ | $F_1$ score ↑ | Coll. Rate ↓ | Dist. Err. ↓ |
| ResNet | 68.5% | 54.0% | 2.91 | 51.8% | 68.1% | 3.60 |
| RTNet | 77.3% | 36.8% | 2.89 | 55.9% | 63.8% | 2.94 |
| L1R32H4 | 11.9% | 100.0% | NA. | 12.0% | 91.5% | 100.0 |
| Ours | **80.2%** | **32.8%** | **2.84** | **58.8%** | **57.8%** | **2.81** |
| SUP | 84.9% | 28.3% | 2.75 | 74.6% | 52.9% | 2.90 |

resulting in a high collision rate. Our method is nearly comparable to the supervised reference model SUP on the Kitti dataset. On the nuScenes dataset, our method even produces slightly less distance error of safe paths than the SUP method.

## 5 Related Work

**Neuro-symbolic learning.** Neuro-symbolic learning has received great attention recently. For instance, Dai et al. [2019] and Corapi et al. [2010] suggest bridging neural perception and logical reasoning via an abductive approach, where a logic program is abstracted from a given knowledge base. To reduce reliance on knowledge bases, Ciravegna et al. [2020] and Dong et al. [2019] directly represent and learn constraints using neural networks. However, the learned constraints are still uninterpretable. To improve interpretability, Wang et al. [2019] introduce SATNet, a method that relaxes the MaxSAT problem with semidefinite programming and incorporates it as a layer into neural networks. SATNet is further followed up by several works [Topan et al., 2021, Lim et al., 2022, Yang et al., 2023]. However, how to explicitly extract and use the learned constraints is still unclear for these works. In contrast to the existing neuro-symbolic learning methods, our method can synthesize explicit logical constraints supporting exact reasoning by off-the-shelf reasoning engines.

**Constraint learning.** Our work is also related to constraint learning, which can be traced back to *Valiant's algorithm* [Valiant, 1984] and, more generally, *inductive logic programming* [Muggleton and De Raedt, 1994, Bratko and Muggleton, 1995, Yang et al., 2017, Evans and Grefenstette, 2018]. However, Cropper and Dumančić [2022] highlight that inductive logic programming is limited when learning from raw data, such as images and speech, as opposed to perfect symbolic data. To this end, our method goes a step further by properly tackling the symbol grounding problem.

**Boolean quadratic programming and its relaxation.** Many constraint learning and logical reasoning tasks, e.g., learning Pseudo-Boolean function [Marichal and Mathonet, 2010], MaxSAT learning [Wang et al., 2019] and solving [Gomes et al., 2006], and SAT solving [Lipp and Boyd, 2016], can be formulated as Boolean quadratic programming (i.e., quadratic programming with binary variables) [Hammer and Rubin, 1970]. However, commonly used techniques, such as branch and bound [Buchheim et al., 2012] and cutting plane [Kelley, 1960], cannot be applied in neuro-symbolic learning tasks. In literature, semidefinite relaxation (SDR) [d'Aspremont and Boyd, 2003, Gomes et al., 2006, Wang and Kolter, 2019] and difference-of-convex (DC) programming [Tao and Hoai An, 1997, Yuille and Rangarajan, 2003, Lipp and Boyd, 2016, Hoai An and Tao, 2018] are two typical methods to relax Boolean constraints. Although SDR is generally more efficient, the tightness and recovering binary results from relaxation are still an open problem [Burer and Ye, 2020, Wang and Kılınç-Karzan, 2022], compromising the exactness of logical reasoning. In this work, we choose DC programming and translate DC constraints to a penalty term with gradually increasing weight, so as to ensure that the Boolean constraints can be finally guaranteed.

## 6 Limitations

In this section, we discuss the limitations of our framework and outline some potential solutions.

**Expressiveness.** The theoretical capability of cardinality constraints to represent any propositional logic formula does not necessarily imply the practical ability to learn any such formula in our frame-

work; this remains a challenge. Fundamentally, logical constraint learning is an inductive method, and thus different learning methods would have different inductive biases. Cardinality constraint-based learning is more suitable for tasks where the logical constraints can be straightforwardly translated into the cardinality form. A typical example of such a task is Sudoku, where the target CNF formula consists of at least 8,829 clauses [Lynce and Ouaknine, 2006], while the total number of target cardinality constraints stands at a mere 324.

Technically, our logical constraint learning prefers equality constraints (e.g., $x + y = 2$), which actually induce logical conjunction (e.g., $x \wedge y = \mathrm{T}$) and may ignore potential logical disjunction which is represented by inequality constraints (e.g., $x \vee y = \mathrm{T}$ is expressed by $x + y \geq 1$). To overcome this issue, a practical trick is to introduce some auxiliary variables, which is commonly used in linear programming [Fang and Puthenpura, 1993]. Consider the disjunction $x \vee y = \mathrm{T}$; here, the auxiliary variables $z_1, z_2$ help form two equalities, namely, $x + y + z_1 = 2$ for $(x, y) = (\mathrm{T}, \mathrm{T})$ and $x + y + z_2 = 1$ for $(x, y) = (\mathrm{T}, \mathrm{F})$ or $(x, y) = (\mathrm{F}, \mathrm{T})$. One can refer to the Chain-XOR task (cf. Section F.1) for a concrete application of auxiliary variables.

**Reasoning efficiency.** The reasoning efficiency, particularly that of SMT solvers, during the inference phase can be a primary bottleneck in our framework. For instance, in the self-driving path planning task, when we scale the map size up to a $20 \times 20$ grid involving $800$ Boolean variables ($400$ variables for grid obstacles and $400$ for path designation), the Z3 MaxSAT solver takes more than two hours for some inputs.

To boost reasoning efficiency, there are several practical methods that could be applied. One straightforward method is to use an integer linear program (ILP) solver (e.g., Gurobi) as an alternative to the Z3 MaxSAT solver. In addition, some learning-based methods (e.g., Balunovic et al. [2018]) may enhance SMT solvers in our framework. Nonetheless, we do not expect that merely using a more efficient solver can resolve the problem. The improve the scalability, a more promising way is to combine System 1 and System 2 also in the inference stage (e.g., Cornelio et al. [2023]). Generally speaking, in the inference stage, neural perception should first deliver a partial solution, which is then completed by the reasoning engine. Such a paradigm ensures fast reasoning via neural perception, drastically reducing the logical variables that need to be solved by the exact reasoning engine, thereby also improving its efficiency.

## 7 Conclusion

This paper presents a neuro-symbolic learning approach that conducts neural network training and logical constraint synthesis simultaneously, fueled by symbol grounding. The gap between neural networks and symbol logic is suitably bridged by cardinality constraint-based learning and difference-of-convex programming. Moreover, we introduce the trust region method to effectively prevent the degeneracy of logical constraint learning. Both theoretical analysis and empirical evaluations have confirmed the effectiveness of the proposed approach. Future work could explore constraint learning using large language models to trim the search space of the involved logical variables, and augment reasoning efficiency by further combining logical reasoning with neural perception.

## Acknowledgment

We are thankful to the anonymous reviewers for their helpful comments. This work is supported by the National Natural Science Foundation of China (Grants #62025202, #62172199). T. Chen is also partially supported by Birkbeck BEI School Project (EFFECT) and an overseas grant of the State Key Laboratory of Novel Software Technology under Grant #KFKT2022A03. Yuan Yao (y.yao@nju.edu.cn) and Xiaoxing Ma (xxm@nju.edu.cn) are the corresponding authors.

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

# A  Proofs of DC technique

**Notations.** We define $\boldsymbol{S} := (\boldsymbol{Q}^\mathsf{T}\boldsymbol{Q} + \tau\boldsymbol{I})$, $\boldsymbol{s} := (\boldsymbol{Q}^\mathsf{T}\boldsymbol{q}_1 + \tau\boldsymbol{q}_2)$, and denote the largest eigenvalues and largest diagonal element of $\boldsymbol{S}$ by $\sigma_{\max}$ and $\delta_{\max}$, respectively. Hence, the two problems can be equivalently rewritten as

$$\text{(P)} \quad \min_{\boldsymbol{u}\in\{0,1\}^n} \boldsymbol{u}^\mathsf{T}\boldsymbol{S}\boldsymbol{u} - 2\boldsymbol{s}^\mathsf{T}\boldsymbol{u}, \qquad \text{(P}_t\text{)} \quad \min_{\boldsymbol{u}\in[0,1]^n} \boldsymbol{u}^\mathsf{T}(\boldsymbol{S} - t\boldsymbol{I})\boldsymbol{u} - (2\boldsymbol{s} - t\boldsymbol{e})^\mathsf{T}\boldsymbol{u}.$$

## A.1  Proof of Proposition 1

*Proof.* The results are primarily based on Bertsekas [2015, Proposition 1.3.4]: the minima of a strictly concave function cannot be in the relative interior of the feasible set.

We first show that if $t_0 \geq \sigma_{\max}$, then the two problems are equivalent [Le Thi and Ding Tao, 2001, Theorem 1]. Specifically, since $\boldsymbol{S} - t\boldsymbol{I}$ is negative definite, problem (P$_t$) is strictly concave. Therefore, the minima should be in the vertex set of the feasible domain, which is consistent with problem (P).

We can further generalize this result to the case $t_0 \geq \delta_{\max}$[Hansen et al., 1993, Proposition 1]. In this case, considering the $i$-th component of $\boldsymbol{u}$, its second-order derivative in problem (P$_t$) is $2(\boldsymbol{S}_{ii} - t)$. Similarly, the strict concavity of $\boldsymbol{u}_i$ ensures a binary solution, indicating the equivalence of problems (P) and (P$_t$). $\qquad\square$

## A.2  Proof of Proposition 2

*Proof.* The Karush–Kuhn–Tucker (KKT) conditions of the problem (P$_t$) are as follows.

$$[2\boldsymbol{S}\boldsymbol{u} - 2t\boldsymbol{u} - 2\boldsymbol{s} + t\boldsymbol{e}]_i - \boldsymbol{\alpha}_i + \boldsymbol{\beta}_i = \boldsymbol{0};$$
$$\boldsymbol{u}_i \in [0,1]^n; \quad \boldsymbol{\alpha}_i \geq \boldsymbol{0}, \boldsymbol{\beta}_i \geq \boldsymbol{0};$$
$$\boldsymbol{\alpha}_i\boldsymbol{u}_i = 0, \quad \boldsymbol{\beta}_i(\boldsymbol{u}_i - 1) = 0; \quad i = 1,\ldots,n.$$

where $\boldsymbol{\alpha}$ and $\boldsymbol{\beta}$ are multiplier vector. For $\boldsymbol{u} \in \{0,1\}^n$, the KKT condition is equivalent to

$$\boldsymbol{\alpha}_i = [2\boldsymbol{S}\boldsymbol{u} - 2t\boldsymbol{u} - 2\boldsymbol{s} + t\boldsymbol{e}]_i(1 - \boldsymbol{u}_i) \geq 0, \quad \boldsymbol{\beta}_i = [2\boldsymbol{S}\boldsymbol{u} - 2t\boldsymbol{u} - 2\boldsymbol{s} + t\boldsymbol{e}]_i\boldsymbol{u}_i \leq 0.$$

By using $(1 - 2\boldsymbol{u}_i) \in \{-1, 1\}$, we can further combine the above two inequalities, and obtain

$$2[\boldsymbol{S}\boldsymbol{u} - \boldsymbol{s}]_i(1 - 2\boldsymbol{u}_i) + t \geq 0, \quad i = 1,\ldots,n.$$

On the other hand, if $2[\boldsymbol{S}\boldsymbol{u} - \boldsymbol{s}]_i(1 - 2\boldsymbol{u}_i) + t \geq 0$ holds for each $i = 1,\ldots,n$, it is easy to check that $\boldsymbol{\alpha} \geq 0$ and $\boldsymbol{\beta} \geq 0$, which proves the first part of the proposition.

The proof of the second part is a direct result of Beck and Teboulle [2000, Theorem 2.4]. To be specific, if $\boldsymbol{u}$ achieves a global minimum of (P), then $q(\boldsymbol{u}) \leq q(\boldsymbol{u}')$ for any $\boldsymbol{u}' \in \{0,1\}^n$. Hence, we only flip the $i$-th value of $\boldsymbol{u}$, i.e., considering $\boldsymbol{u}_i$ and $\boldsymbol{u}_i' = 1 - \boldsymbol{u}_i$, and it holds that

$$\boldsymbol{u}^\mathsf{T}\boldsymbol{S}\boldsymbol{u} - 2\boldsymbol{s}^\mathsf{T}\boldsymbol{u} \leq (\boldsymbol{u}')^\mathsf{T}\boldsymbol{S}\boldsymbol{u}' - 2\boldsymbol{s}^\mathsf{T}\boldsymbol{u}'$$
$$= (\boldsymbol{u}^\mathsf{T}\boldsymbol{S}\boldsymbol{u} - 2\boldsymbol{s}^\mathsf{T}\boldsymbol{u}) + 2[\boldsymbol{S}\boldsymbol{u} - \boldsymbol{s}]_i(1 - 2\boldsymbol{u}_i) + \boldsymbol{S}_{ii}.$$

Rearranging the inequality, we obtain

$$2[\boldsymbol{S}\boldsymbol{u} - \boldsymbol{s}]_i(1 - 2\boldsymbol{u}_i) \geq -\boldsymbol{S}_{ii}, \quad i = 1,\ldots,n,$$

which completes the proof. $\qquad\square$

# B  Proof of Theorem 1

*Proof.* **Notations.** We use $\|\cdot\|$ to denote the $\ell_2$ norm for vectors and Frobenius norm for matrices. We define

$$\varphi(\boldsymbol{\phi},\boldsymbol{\theta},\mathbf{Z},\mathbf{Y}) := \|\mathbf{Z}\boldsymbol{w}_u + \mathbf{Y}\boldsymbol{w}_v - \boldsymbol{b}\|^2 + \alpha\|(\mathbf{Z},\mathbf{Y}) - (f_{\boldsymbol{\theta}}(\mathbf{X}),\mathbf{Y})\|^2 + \lambda\|\boldsymbol{w} - \boldsymbol{w}^0\|^2.$$

For the loss functions of logic programming and network training, we assume $\ell_1(\boldsymbol{\theta})$ and $\ell_2(\boldsymbol{\phi})$ to be $\mu_{\boldsymbol{\theta}}$ and $\mu_{\boldsymbol{\phi}}$ smooth, respectively. For ease of presentation, we define $\Delta^k = f_{\boldsymbol{\theta}^k}(\mathbf{X})\boldsymbol{w}_u^k + \mathbf{Y}\boldsymbol{w}_v^k - \boldsymbol{b}$,

and let $c_{\max}$ be the upper bound of $\|\Delta^k\|$. Furthermore, by using the Woodbury identity formula, we can compute

$$(\mathbf{Z}^k; \mathbf{Y}^k) = \arg\min_{(\mathbf{Z},\mathbf{Y})} \|\mathbf{Z}\boldsymbol{w}_u^k + \mathbf{Y}\boldsymbol{w}_v^k - \boldsymbol{b}\|^2 + \alpha\|(\mathbf{Z},\mathbf{Y}) - (f_{\boldsymbol{\theta}^k}(\mathbf{X}), \mathbf{Y})\|^2 + \lambda\|\boldsymbol{w} - \boldsymbol{w}^0\|^2$$

$$= (f_{\boldsymbol{\theta}^k}(\mathbf{X}); \mathbf{Y}) - \beta^k \Delta^k (\boldsymbol{w}^k)^{\mathsf{T}}, \quad \text{where} \quad \beta^k = \frac{1}{\alpha + \|\boldsymbol{w}^k\|^2}.$$

Let $\rho^k := (\alpha\beta^k)$, we have

$$\varphi(\boldsymbol{\phi}^k, \boldsymbol{\theta}^k, f_{\boldsymbol{\theta}^k}(\mathbf{X}), \mathbf{Y}) - \varphi(\boldsymbol{\phi}^k, \boldsymbol{\theta}^k, \mathbf{Z}^k, \mathbf{Y}^k) = (1 - ((\alpha\beta^k)^2 + (1 - \alpha\beta^k)^2)\|\Delta^k\|^2$$
$$= 2\rho^k(1 - \rho^k)\|\Delta^k\|^2.$$

**Update of $\phi$.** We consider the single rule case (multiple rules can be directly decomposed), i.e., $\phi = (\boldsymbol{w}, \boldsymbol{b})$ and $\boldsymbol{b} = (b; \ldots; b)$. The update of $\phi$ is conducted on the loss function

$$\ell_2^k(\boldsymbol{w}, \boldsymbol{b}) = \varphi(\boldsymbol{\phi}^k, \boldsymbol{\theta}^k, f_{\boldsymbol{\theta}^k}(\mathbf{X}), \mathbf{Y}) = \|f_{\boldsymbol{\theta}^k}(\mathbf{X})\boldsymbol{w}_u + \mathbf{Y}\boldsymbol{w}_v - \boldsymbol{b}\|^2 + \lambda\|\boldsymbol{w} - \boldsymbol{w}^0\|^2.$$

The smallest and the largest eigenvalues of $(f_{\boldsymbol{\theta}^k}(\mathbf{X}), \mathbf{Y})^{\mathsf{T}}(f_{\boldsymbol{\theta}^k}(\mathbf{X}), \mathbf{Y}) + \lambda\boldsymbol{I}$ are denoted by $\sigma_{\min}$ and $\sigma_{\max}$, respectively.

The PPA method updates $\boldsymbol{w}$ by

$$\boldsymbol{w}^{k+1} = \arg\min_{\boldsymbol{w}} \ell_2^k(\boldsymbol{w}, \boldsymbol{b}) + \frac{1}{\gamma}\|\boldsymbol{w} - \boldsymbol{w}^k\|^2,$$

which can be reduced to

$$\boldsymbol{w}^{k+1} - \boldsymbol{w}^k = -\boldsymbol{M}^k\boldsymbol{\delta}^k, \quad \boldsymbol{\delta}^k = (f_{\boldsymbol{\theta}^k}(\mathbf{X}), \mathbf{Y})^{\mathsf{T}}\Delta = \nabla_{\boldsymbol{w}}\ell_2^k(\boldsymbol{w}, \boldsymbol{b}),$$

where

$$\boldsymbol{M}^k = \big((f_{\boldsymbol{\theta}^k}(\mathbf{X}), \mathbf{Y})^{\mathsf{T}}(f_{\boldsymbol{\theta}^k}(\mathbf{X}), \mathbf{Y}) + \lambda\boldsymbol{I} + \frac{1}{\gamma}\boldsymbol{I}\big)^{-1}.$$

The $(2/\gamma)$-strongly convexity of the proximal term implies the Polyak-Łojasiewicz (PL) inequality, which derives that

$$\varphi(\boldsymbol{\phi}^k, \boldsymbol{\theta}^k, \mathbf{Z}^k, \mathbf{Y}^k) = \varphi(\boldsymbol{\phi}^k, \boldsymbol{\theta}^k, f_{\boldsymbol{\theta}^k}(\mathbf{X}), \mathbf{Y}) - 2\rho^k(1 - \rho^k)\|\Delta\|^2$$
$$= \ell_2^k(\boldsymbol{w}^k, \boldsymbol{b}) - 2\rho^k(1 - \rho^k)\|\Delta^k\|^2 \geq \ell_2^k(\boldsymbol{w}^{k+1}, \boldsymbol{b}) - 2\rho^k(1 - \rho^k)\|\Delta\|^2 + \frac{2}{\gamma}\|\boldsymbol{w}^{k+1} - \boldsymbol{w}^k\|^2.$$

Plugging $\boldsymbol{w}^{k+1} - \boldsymbol{w}^k = -\boldsymbol{M}^k\boldsymbol{\delta}^k$ into the inequality, we have

$$\varphi(\boldsymbol{\phi}^k, \boldsymbol{\theta}^k, \mathbf{Z}^k, \mathbf{Y}^k) \geq \ell_2^k(\boldsymbol{w}^{k+1}, \boldsymbol{b}) + \frac{2}{\gamma}(\boldsymbol{\delta}^k)^{\mathsf{T}}(\boldsymbol{M}^k)^2\boldsymbol{\delta}^k - 2\rho^k(1 - \rho^k)\|\Delta^k\|^2$$

$$\geq \varphi(\boldsymbol{\phi}^{k+1}, \boldsymbol{\theta}^k, \mathbf{Z}^{k+\frac{1}{2}}, \mathbf{Y}^{k+\frac{1}{2}}) + \frac{2}{\gamma}(\boldsymbol{\delta}^k)^{\mathsf{T}}(\boldsymbol{M}^k)^2\boldsymbol{\delta}^k - 2\rho^k(1 - \rho^k)\|\Delta^k\|^2,$$

where

$$(\mathbf{Z}^{k+\frac{1}{2}}; \mathbf{Y}^{k+\frac{1}{2}}) = \arg\min_{(\bar{\mathbf{Z}},\bar{\mathbf{Y}})} \|\bar{\mathbf{Z}}\boldsymbol{w}_u^{k+1} + \bar{\mathbf{Y}}\boldsymbol{w}_v^{k+1} - \boldsymbol{b}\|^2 + \alpha\|(\bar{\mathbf{Z}}, \bar{\mathbf{Y}}) - (f_{\boldsymbol{\theta}^k}(\mathbf{X}), \mathbf{Y})\|^2$$

$$= (f_{\boldsymbol{\theta}^k}(\mathbf{X}); \mathbf{Y}) - \beta^{k+\frac{1}{2}}\Delta^{k+\frac{1}{2}}(\boldsymbol{w}^{k+1})^{\mathsf{T}}, \quad \text{where} \quad \beta^{k+\frac{1}{2}} = \frac{1}{\alpha + \|\boldsymbol{w}^{k+1}\|^2}.$$

Note that $(\boldsymbol{M}^k)^2$ has the smallest eigenvalue $\gamma^2/(1 + \gamma\sigma_{\max})^2$, and thus we have

$$\varphi(\boldsymbol{\phi}^k, \boldsymbol{\theta}^k, \mathbf{Z}^k, \mathbf{Y}^k) \geq \varphi(\boldsymbol{\phi}^{k+1}, \boldsymbol{\theta}^k, \mathbf{Z}^{k+\frac{1}{2}}, \mathbf{Y}^{k+\frac{1}{2}}) + \frac{2\gamma}{(1 + \gamma\sigma_{\max})^2}\|\nabla_{\boldsymbol{\phi}}\ell_2(\boldsymbol{w}, \boldsymbol{b})\|^2 - 2\rho^k(1 - \rho^k)c_{\max}.$$

**Update of $\theta$.** The update of $\theta$ is conducted on the loss function

$$\ell_1^k(\boldsymbol{\theta}) = \|\mathbf{Z}^{k+\frac{1}{2}} - f_{\boldsymbol{\theta}}(\mathbf{X})\|^2.$$

By using $\mu_{\boldsymbol{\theta}}$-smooth of $\ell_1^k$, we obtain that

$$\varphi(\boldsymbol{\phi}^{k+1}, \boldsymbol{\theta}^k, \mathbf{Z}^{k+\frac{1}{2}}, \mathbf{Y}^{k+\frac{1}{2}}) - \varphi(\boldsymbol{\phi}^{k+1}, \boldsymbol{\theta}^{k+1}, \mathbf{Z}^{k+\frac{1}{2}}, \mathbf{Y}^{k+\frac{1}{2}}) = \ell_1^k(\boldsymbol{\theta}^{k+1}) - \ell_1^k(\boldsymbol{\theta}^k)$$

$$\geq -\langle \nabla_{\boldsymbol{\theta}} \ell_1^k(\boldsymbol{\theta}^k), \boldsymbol{\theta}^{k+1} - \boldsymbol{\theta}^k \rangle - \frac{\mu_{\boldsymbol{\theta}}}{2} \|\boldsymbol{\theta}^{k+1} - \boldsymbol{\theta}^k\|^2 \geq \frac{1}{2}\eta \|\nabla_{\boldsymbol{\theta}} \ell_1^k(\boldsymbol{\theta}^k)\|^2.$$

Letting $\mathbf{Z}^{k+1} = \arg\min_{\mathbf{Z}} \varphi(\boldsymbol{\phi}^{k+1}, \boldsymbol{\theta}^{k+1}, \mathbf{Z})$, we conclude

$$\varphi(\boldsymbol{\phi}^{k+1}, \boldsymbol{\theta}^k, \mathbf{Z}^{k+\frac{1}{2}}, \mathbf{Y}^{k+\frac{1}{2}}) \geq \varphi(\boldsymbol{\phi}^{k+1}, \boldsymbol{\theta}^{k+1}, \mathbf{Z}^{k+1}, \mathbf{Y}^{k+1}) + \frac{1}{2}\eta \|\nabla_{\boldsymbol{\theta}} \ell_1^k(\boldsymbol{\theta}^k)\|^2.$$

**Convergent result.** By combining the update of $\boldsymbol{\phi}$ and $\boldsymbol{\theta}$, we have

$$\varphi(\boldsymbol{\phi}^k, \boldsymbol{\theta}^k, \mathbf{Z}^k, \mathbf{Y}^k) - \varphi(\boldsymbol{\phi}^{k+1}, \boldsymbol{\theta}^{k+1}, \mathbf{Z}^{k+1}, \mathbf{Y}^{k+1})$$

$$\geq \frac{1}{2}\eta \|\nabla_{\boldsymbol{\theta}} \ell_1^k(\boldsymbol{\theta}^k)\|^2 + \frac{2\gamma}{(1 + \gamma\sigma_{\max})^2} \|\nabla_{\boldsymbol{\phi}} \ell_2(\boldsymbol{\phi}^k)\|^2 - 2\rho^k(1 - \rho^k)c_{\max}.$$

Taking a telescopic sum over $k$, we obtain

$$\varphi(\boldsymbol{\phi}^0, \boldsymbol{\theta}^0, \mathbf{Z}^0, \mathbf{Y}^0) - \varphi(\boldsymbol{\phi}^K, \boldsymbol{\theta}^K, \mathbf{Z}^K, \mathbf{Y}^K)$$

$$\geq \sum_{i=1}^K \frac{1}{2}\eta \|\nabla_{\boldsymbol{\theta}} \ell_1^k(\boldsymbol{\theta}^k)\|^2 + \frac{2\gamma}{(1 + \gamma\sigma_{\max})^2} \|\nabla_{\boldsymbol{\phi}} \ell_2(\boldsymbol{\phi}^k)\|^2 - 2\rho^k(1 - \rho^k)c_{\max}.$$

Since $\rho^k(1 - \rho^k) \leq \kappa_\rho/(K+1)^2$, we have

$$\mathbb{E}[\|\nabla_{\boldsymbol{\theta}} \ell_2^k(\boldsymbol{\theta}^k)\|^2] \leq \frac{2}{(K+1)\eta} \big((\varphi(\boldsymbol{\phi}^0, \boldsymbol{\theta}^0, \mathbf{Z}^0, \mathbf{Y}^0) - \min\varphi) + 2\kappa c_{\max}\big),$$

and

$$\mathbb{E}[\|\nabla_{\boldsymbol{\phi}} \ell_2(\boldsymbol{\phi}^k)\|^2] \leq \frac{(1 + \gamma\sigma_{\max})^2}{2(K+1)} \big((\varphi(\boldsymbol{\phi}^0, \boldsymbol{\theta}^0, \mathbf{Z}^0, \mathbf{Y}^0) - \min\varphi) + 2\kappa c_{\max}\big).$$

**Stochastic version.** We first introduce an additional assumption: the gradient estimate is unbiased and has bounded variance [Bottou et al., 2018, Sec. 4], i.e.,

$$\mathbb{E}_\xi[\tilde{\nabla}_{\boldsymbol{\theta}} \ell_1^k(\boldsymbol{\theta}^k)] = \nabla_{\boldsymbol{\theta}} \ell_1^k(\boldsymbol{\theta}^k), \quad \mathbb{E}_\xi[\tilde{\nabla}_{\boldsymbol{\theta}} \ell_2^k(\boldsymbol{\phi}^k)] = \nabla_{\boldsymbol{\theta}} \ell_2^k(\boldsymbol{\phi}^k),$$

and

$$\mathbb{V}_\xi[\tilde{\nabla}_{\boldsymbol{\theta}} \ell_1^k(\boldsymbol{\theta}^k)] \leq \zeta + \zeta_v \|\nabla_{\boldsymbol{\theta}} \ell_1^k(\boldsymbol{\theta}^k)\|^2, \quad \mathbb{V}_\xi[\tilde{\nabla}_{\boldsymbol{\phi}} \ell_1^k(\boldsymbol{\theta}^k)] \leq \zeta + \zeta_v \|\nabla_{\boldsymbol{\phi}} \ell_2^k(\boldsymbol{\phi}^k)\|^2.$$

This assumption derives the following inequalities hold for $\zeta_g = \zeta_v + 1$:

$$\mathbb{E}_\xi[\|\tilde{\nabla}_{\boldsymbol{\theta}} \ell_1^k(\boldsymbol{\theta}^k)\|^2] \leq \zeta + \zeta_g \|\nabla_{\boldsymbol{\theta}} \ell_1^k(\boldsymbol{\theta}^k)\|^2, \quad \mathbb{E}_\xi[\|\tilde{\nabla}_{\boldsymbol{\theta}} \ell_2^k(\boldsymbol{\phi}^k)\|^2] \leq \zeta + \zeta_g \|\nabla_{\boldsymbol{\phi}} \ell_2^k(\boldsymbol{\phi}^k)\|^2,$$

For the update of $\boldsymbol{\theta}$, we have

$$\varphi(\boldsymbol{\phi}^{k+1}, \boldsymbol{\theta}^k, \mathbf{Z}^{k+\frac{1}{2}}, \mathbf{Y}^{k+\frac{1}{2}}) - \mathbb{E}_\xi[\varphi(\boldsymbol{\phi}^{k+1}, \boldsymbol{\theta}^{k+1}, \mathbf{Z}^{k+1}, \mathbf{Y}^{k+1})] \geq \frac{\eta_k}{2} \|\nabla_{\boldsymbol{\theta}} \ell_1^k(\boldsymbol{\theta}^k)\|^2 - \frac{\eta_k^2 \mu_{\boldsymbol{\theta}}}{2}\zeta.$$

For the update of $\boldsymbol{\phi}$, using the $\mu_{\boldsymbol{\phi}}$-smooth, and taking the total expectation:

$$\varphi(\boldsymbol{\phi}^k, \boldsymbol{\theta}^k, \mathbf{Z}^k, \mathbf{Y}^k) - \mathbb{E}_\xi[\varphi(\boldsymbol{\phi}^{k+1}, \boldsymbol{\theta}^k, \mathbf{Z}^{k+\frac{1}{2}}, \mathbf{Y}^{k+\frac{1}{2}})] + 2\rho^k(1 - \rho^k)\|\Delta^k\|^2$$

$$\geq (\nabla_{\boldsymbol{\phi}} \ell_2^k(\boldsymbol{\phi}^k))^{\mathsf{T}} \boldsymbol{M}^k (\nabla_{\boldsymbol{\phi}} \ell_2^k(\boldsymbol{\phi}^k)) - \frac{\mu_{\boldsymbol{\phi}}}{2} \mathbb{E}_\xi[\|\tilde{\boldsymbol{M}}^k \tilde{\nabla}_{\boldsymbol{\phi}} \ell_2^k(\boldsymbol{\phi}^k)\|^2]$$

$$\geq \frac{1}{\epsilon^k + \sigma_{\max}} \|\nabla_{\boldsymbol{\phi}} \ell_2^k(\boldsymbol{\phi}^k)\|^2 - \frac{\mu_{\boldsymbol{\phi}}}{2(\epsilon^k + \sigma_{\min})^2}(\zeta + \zeta_g \|\nabla_{\boldsymbol{\phi}} \ell_2^k(\boldsymbol{\phi}^k)\|^2),$$

where we define $\epsilon^k = 1/\gamma^k$ for simplicity. Now, let $\gamma$ be sufficiently small (that is, satisfying $(\epsilon^k + \sigma_{\min})^2 \geq \mu_{\boldsymbol{\phi}}(\epsilon^k + \sigma_{\max})$), we obtain

$$\varphi(\boldsymbol{\phi}^k, \boldsymbol{\theta}^k, \mathbf{Z}^k, \mathbf{Y}^k) - \mathbb{E}_\xi[\varphi(\boldsymbol{\phi}^{k+1}, \boldsymbol{\theta}^k, \mathbf{Z}^{k+\frac{1}{2}}, \mathbf{Y}^{k+\frac{1}{2}})] + 2\rho^k(1 - \rho^k)\|\Delta^k\|^2$$

$$\geq \frac{1}{2(\epsilon^k + \sigma_{\max})} \|\nabla_{\boldsymbol{\phi}} \ell_2^k(\boldsymbol{\phi}^k)\|^2 - \frac{\mu_{\boldsymbol{\phi}}}{2(\epsilon^k + \sigma_{\min})^2}\zeta.$$

Putting the updates of $\boldsymbol{\theta}$ and $\boldsymbol{\phi}$ together, we have

$$\varphi(\boldsymbol{\phi}^k, \boldsymbol{\theta}^k, \mathbf{Z}^k, \mathbf{Y}^k) - \mathbb{E}_\xi[\varphi(\boldsymbol{\phi}^{k+1}, \boldsymbol{\theta}^{k+1}, \mathbf{Z}^{k+1}, \mathbf{Y}^{k+1})] + 2\rho^k(1-\rho^k)\|\Delta^k\|^2$$

$$\geq \frac{1}{2}\eta_k\|\nabla_{\boldsymbol{\theta}}\ell_1^k(\boldsymbol{\theta}^k)\|^2 - \frac{1}{2}\eta_k^2\mu_{\boldsymbol{\theta}}\zeta + \frac{1}{2(\epsilon^k + \sigma_{\max})}\|\nabla_{\boldsymbol{\phi}}\ell_2^k(\boldsymbol{\phi}^k)\|^2 - \frac{\mu_{\boldsymbol{\phi}}}{2(\epsilon^k + \sigma_{\min})^2}\zeta.$$

Now, setting $\eta^k \leq \kappa_{\boldsymbol{\theta}}/\sqrt{K+1}$ and $\gamma^k \leq \kappa_{\boldsymbol{\phi}}/\sqrt{K+1}$, we can conclude

$$\mathbb{E}[\|\nabla_{\boldsymbol{\theta}}\ell_1^k(\boldsymbol{\theta}^k)\|^2] = \mathcal{O}(\frac{1}{\sqrt{K+1}}), \quad \mathbb{E}[\|\nabla_{\boldsymbol{\phi}}\ell_2(\boldsymbol{\phi}^k)\|^2] = \mathcal{O}(\frac{1}{\sqrt{K+1}}). \qquad \square$$

## C  Proof of Theorem 2

*Proof.* We consider the following problem,

$$(\mathrm{P}_\xi) \quad \min_{\boldsymbol{u}\in\{0,1\}^n} q_\xi(\boldsymbol{u}) := \boldsymbol{u}^\top(\boldsymbol{S}+\lambda\boldsymbol{I})\boldsymbol{u} - 2(\boldsymbol{s}+\lambda\boldsymbol{\xi})^\top\boldsymbol{u}.$$

For given $t \geq 0$, the corresponding stationary points of $(\mathrm{P}_\xi)$ satisfy

$$2[\boldsymbol{S}\boldsymbol{u}-\boldsymbol{s}]_i(1-2\boldsymbol{u}_i) + 2\lambda(\boldsymbol{u}_i-\boldsymbol{\xi}_i)(1-2\boldsymbol{u}_i) + t \geq 0, i = 1, \ldots, n.$$

Note that

$$(\boldsymbol{u}_i-\boldsymbol{\xi}_i)(1-2\boldsymbol{u}_i) = \left\{ \begin{array}{ll} -\boldsymbol{\xi}_i & \text{if} \quad \boldsymbol{u}_i = 0; \\ \boldsymbol{\xi}_i - 1 & \text{if} \quad \boldsymbol{u}_i = 1. \end{array} \right.$$

For given $\boldsymbol{u} \in \{0,1\}^n$, we denote $\varrho_i = 2[\boldsymbol{S}\boldsymbol{u}-\boldsymbol{s}]_i(1-2\boldsymbol{u}_i)$. Then, the probability that $(\mathrm{P}_\xi)$ has the stationary point $\boldsymbol{u}$ can be computed as

$$\Pr(\boldsymbol{u}) = \prod_{i=1}^n \Pr(\varrho_i + 2\lambda(\boldsymbol{u}_i-\boldsymbol{\xi}_i)(1-2\boldsymbol{u}_i) + t \geq 0),$$

where

$$\Pr(2\lambda(\boldsymbol{u}_i-\boldsymbol{\xi}_i)(1-2\boldsymbol{u}_i) + \varrho_i + t \geq 0) = \min(\frac{1}{2\lambda}(t+\varrho_i), 1).$$

Hence, for given two different $\boldsymbol{u}_1^{(0)}$ and $\boldsymbol{u}_2^{(0)}$, the probability that the corresponding rules can converge to the same result $\boldsymbol{u}$ satisfying

$$\Pr(\boldsymbol{u}_1 = \boldsymbol{u}, \boldsymbol{u}_2 = \boldsymbol{u}) \leq \Pr(\boldsymbol{u})^2 = \prod_{i=1}^n \min(\frac{1}{2\lambda}(t+\varrho_i), 1)^2. \qquad \square$$

## D  Trust Region Method

Figure 3 illustrates the key concept of the trust region method. For simplicity, centre points $\boldsymbol{w}_1(\boldsymbol{0}), \ldots, \boldsymbol{w}_4(\boldsymbol{0})$ of the trust region are also set as the initial points of stochastic gradient descent. Stochastic gradient descent is implicitly biased to least norm solutions and finally converges to point $(0, 1)$ by enforcing the Boolean constraints. The trust region penalty encourages the stochastic gradient descent to converge to different optimal solutions in different trust regions.

## E  Experiment Details

**Computing configuration.** We implemented our approach via the PyTorch DL framework. The experiments were conducted on a GPU server with two Intel Xeon Gold 5118 CPU@2.30GHz, 400GB RAM, and 9 GeForce RTX 2080 Ti GPUs. The server ran Ubuntu 16.04 with GNU/Linux kernel 4.4.0.

**Hyperparameter tuning.** Some hyperparameters are introduced in our framework. In Table 3 we summarize the (hyper-)parameters, together with their corresponding initialization or update strategies. Most of these hyperparameters are quite stable and thus only need to be fixed to a constant or set by standard strategies. We only discuss the selection of $m$, and the setting of $\boldsymbol{b}_{\min}, \boldsymbol{b}_{\max}$ and $\boldsymbol{b}$.

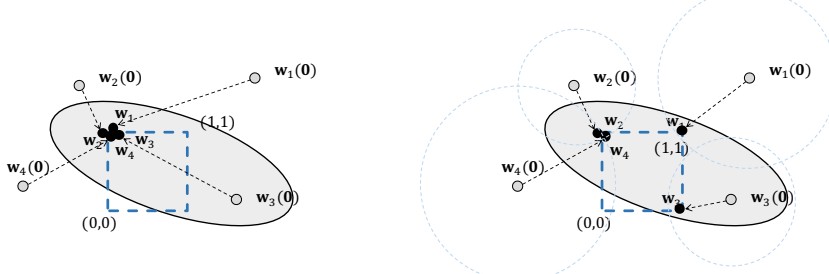

Figure 3: Avoid degeneracy by trust region method. In logical constraint learning, the imposition of the Boolean constraints and the implicit bias of the stochastic gradient descent cause $w_1, \ldots, w_4$ to converge to the same result (left figure), while the trust region constraints guarantee that they can sufficiently indicate different rules (right figure).

(1) To ensure the sufficiency of learned constraints, we suggest initially setting a large $m$ to estimate the actual number of logical constraints needed, and then adjusting it for more efficient training. We also observe that a large $m$ does not ruins the performance of our method. For example, we set $m = 2000$ in visual SudoKu solving task, while only $324$ constraints are learned. (2) For the bias term $b$, we recommend $b$ to be tuned manually rather than set by PPA update, and one can gradually increase $b$ from $1$ to $n - 1$ ($n$ is the number of involved logical variables), and collect all logical constraints as candidate constraints. For $b_{\min}$ and $b_{\max}$, due to the prediction error, it is unreasonable to set $b_{\min}$ and $b_{\max}$ that ensure all examples to satisfy the logical constraint. An alternative method is to set a threshold (e.g. $k\%$) on the training (or validation) set, and the constraint is only required to be satisfied by at least $k\%$ examples.

Table 3: The list of (hyper-)parameters and their initialization or update strategies.

| Param. | Description | Setting |
|---|---|---|
| $\theta$ | Neural network parameters | Updated by stochastic gradient descent |
| $W$ | Matrix of logical constraints | Updated by stochastic PPA |
| $b$ | Bias term of logical constraints | Pre-set or Updated by stochastic PPA |
| $b_{\min}/b_{\max}$ | Lower/Upper bound of logical constraints | Estimated by training set |
| $m$ | Pre-set number of constraints | Adaptively tuned |
| $\alpha$ | Trade-off weight in symbol grounding | Fixed to $\alpha = 0.5$ |
| $\lambda$ | Weight of trust region penalty | Fixed to $\lambda = 0.1$ |
| $t_1/t_2$ | Weight of DC penalty | Increased per epoch |
| $\eta$ | Learning rate of network training | Adam schedule |
| $\gamma$ | Step size of constraint learning | Adaptively set ($\gamma = 0.001$ by default) |

## F  Additional Experiment Results

### F.1  Chained XOR

The chained XOR, also known as the parity function, is a basic logical function, yet it has proven challenging for neural networks to learn it explicitly [Shalev-Shwartz et al., 2017, Wang et al., 2019] To be specific, given a sequence of length $L$, the parity function outputs $1$ if there are an odd number of $1$'s in the sequence, and $0$ otherwise. The goal of the Chained XOR task is to learn this parity function with fixed $L$. Note that this task does not involve any perception task.

We compare our method with SATNet and L1R32H4. In this task, SATNet uses an implicit but strong background knowledge that the task can be decomposed into $L$ single XOR tasks. Neither L1R32H4

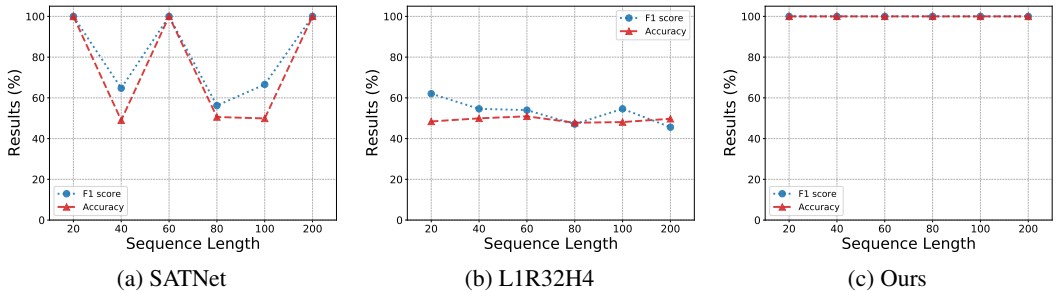

Figure 4: Results (%) of chained XOR task, including accuracy and $F_1$ score (of class 0). The sequence length ranges from 20 to 200, showing that our method stably outperforms competitors.

nor our method uses such knowledge. For L1R32H4, we adapt the embedding layer to this task and fix any other configuration. Regarding our method, we introduce $L - 1$ auxiliary variables.[7]

It is worth noting that these auxiliary variables essentially serve as a form of symbol grounding. Elaborately, the learned logical constraints by our method can be formulated as follows,

$$\boldsymbol{w}_1\mathbf{x}_1 + \cdots + \boldsymbol{w}_L\mathbf{x}_L + \boldsymbol{w}_{L+1}\mathbf{z}_1 + \cdots + \boldsymbol{w}_{2L-1}\mathbf{z}_{L-1} = b,$$

where $\boldsymbol{w}_i \in \mathcal{B}, i = 1, \ldots, 2L-1$, $\mathbf{x}_i \in \mathcal{B}, i = 1, \ldots, L$ and $\mathbf{z}_i \in \mathcal{B}, i = 1, \ldots, L$. The auxiliary variables $\mathbf{z}_i, i = 1, \ldots, L$ have different truth assignments for different examples, indicting *how* the logical constraint is satisfied by the given input. Now, combining the symbol grounding of auxiliary variables, we revise the optimization problem (1) of our framework as

$$\min_{(\boldsymbol{W}, \boldsymbol{b})} \mathbb{E}_{(\mathbf{x}, \mathbf{y}) \sim \mathcal{D}}[\|\boldsymbol{W}(\mathbf{x}; \bar{\mathbf{z}}; \mathbf{y}) - \boldsymbol{b}\|^2] + \lambda \|\boldsymbol{W} - \boldsymbol{W}^{(0)}\|^2,$$

$$\text{s.t.} \quad \bar{\mathbf{z}} = \arg\min_{\mathbf{z} \in \mathcal{Z}} \mathbb{E}_{(\mathbf{x}, \mathbf{y}) \sim \mathcal{D}}[\|\boldsymbol{W}(\mathbf{x}; \mathbf{z}; \mathbf{y}) - \boldsymbol{b}\|^2], \quad \boldsymbol{W} \in \mathcal{B}^{m \times (u+v)}, \quad \boldsymbol{b} \in \mathcal{N}_+^m.$$

The symbol grounding is solely guided by logical constraints, as neural perception is not involved.

The experimental results are plotted in Figure 4. The results show that L1R32H4 is unable to learn such a simple reasoning pattern, while SATNet often fails to converge even with sufficient iterations, leading to unstable results. Our method consistently delivers full accuracy across all settings, thereby demonstrating superior performance and enhanced scalability in comparison to existing state-of-the-art methods. To further exemplify the efficacy of our method, we formulate the learned constraints in the task of $L = 20$. Eliminating redundant constraints and replacing the auxiliary variables with logical disjunctions, the final learned constraint can be expressed as

$$(\mathbf{x}_1 + \cdots + \mathbf{x}_{20} + y = 0) \vee (\mathbf{x}_1 + \cdots + \mathbf{x}_{20} + y = 2) \vee \cdots \vee (\mathbf{x}_1 + \cdots + \mathbf{x}_{20} + y = 20),$$

which shows that our method concludes with complete and precise logical constraints.

### F.2 Nonograms

Nonograms is a logic puzzle with simple rules but challenging solutions. Given a grid of squares, the task of nonograms is to plot a binary image, i.e., filling each grid in black or marking it by X. The required numbers of black squares on that row (resp. column) are given beside (resp. above) each row (resp. column) of the grid. Figure 5 gives a simple example.

In contrast to the supervised setting used in Yang et al. [2023], we evaluate our method on a weakly supervised learning setting. Elaborately, instead of the fully solved board, only partial solutions (i.e., only one row or one column) are observed. Note that this supervision is enough to solve the nonograms, because the only logical rule to be learned is that the different black squares (in each row or column) should not be connected.

For our method, we do not introduce a neural network in this task, and only aim to learn the logical constraints. We carry out the experiments on $7 \times 7$ nonograms, with training data sizes ranging

---

[7]Note that the number of auxiliary variables should not exceed the number of logical variables. If so, the logical constraints trivially converge to any result.

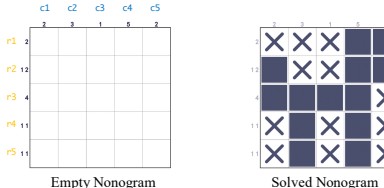

Empty Nonogram    Solved Nonogram

Figure 5: An example of nonograms.

| Data Size | L1R32H4 | Ours |
|-----------|---------|------|
| 1000 | 14.4 | **100.0** |
| 5000 | 62.0 | **100.0** |
| 9000 | 81.2 | **100.0** |

Table 4: Accuracy (%) of the nonograms task.

from 1,000 to 9,000. The results are given in Table 4, showing the efficacy of our logical constraint learning. Compared to the L1R32H4 method, whose effectiveness highly depends on the training data size, our method works well even with extremely limited data.

### F.3 Visual SudoKu Solving

In the visual SudoKu task, it is worth noting that the computation of $\mathbf{z}$ cannot be conducted by batch processing. This is because the index of $\mathbf{y}$ varies for each data point. For instance, in different SudoKu games, the cells to be filled are different, and thus the symbol $\mathbf{z}$ has to be computed in a point-wise way. To solve this issue, we introduce an auxiliary $\bar{\mathbf{y}}$ to approximate the output symbol $\mathbf{y}$:

$$(\bar{\mathbf{z}}, \bar{\mathbf{y}}) = \arg\min_{\bar{\mathbf{z}} \in \mathcal{Z}, \bar{\mathbf{y}} \in \mathcal{Y}} \|\boldsymbol{W}(\bar{\mathbf{z}}; \bar{\mathbf{y}}) - \boldsymbol{b}\|^2 + \alpha \|(\bar{\mathbf{z}}; \bar{\mathbf{y}}) - (f_{\boldsymbol{\theta}}(\mathbf{x}); \mathbf{y})\|^2.$$

On the SATNet dataset, we use the recurrent transformer as the perception model [Yang et al., 2023], because we observe that the recurrent transformer can significantly improve the perception accuracy, and even outperforms the state-of-the-art of MNIST digit recognition model. However, we find that its performance degrades on the more difficult dataset RRN, and thus we still use a standard convolutional neural network model as the perception model for this dataset.

We include detailed results of board and cell accuracy in Table 5. It can be observed that our method is consistently superior to the existing methods, and significantly outperforms the current state-of-the-art method L1R32H4 on the RRN dataset (total board accuracy improvement exceeds 20%). Also note that the solving accuracy of our method always performs the best, illustrating the efficacy of our logical constraint learning.

Next, we exchange the evaluation dataset, namely, using the RRN dataset to evaluate the model trained on the SATNet dataset, and vice versa. The results are presented in Table 6. The accurate logical constraints and exact logical reasoning engine guarantee the best performance of our method on transfer tasks. Notably, the performance of L1R32H4 drops significantly when transferring the model (trained SATNet dataset) to RNN dataset, our method remains unaffected by such shift.

### F.4 Self-driving Path Planning

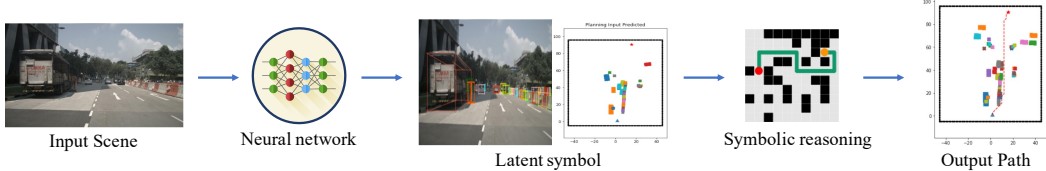

Input Scene    Neural network    Latent symbol    Symbolic reasoning    Output Path

Figure 6: A neuro-symbolic system in self-driving tasks. The neural perception detects the obstacles from the image collected by the camera; the symbolic reasoning plans the driving path based on the obstacle map. The neuro-symbolic learning task is to build these two modules in an end-to-end way.

The goal of the self-driving path planning task is to train the neural network for object detection and to learn the logical constraints for path planning in an end-to-end way. As shown in Figure 6, we construct two maps and each contains $10 \times 10$ grids (binary variables). The neural perception detects the obstacles from the image $\mathbf{x}$ and locates it in the first map, which is essentially the symbol $\mathbf{z}$. Next,

Table 5: Detailed cell and board accuracy (%) of **original** visual Sudoku task.

| Method | SATNet dataset | | | RRN dataset | | |
|---|---|---|---|---|---|---|
| | Perception board acc. | Solving board acc. | Total board acc. | Perception board acc. | Solving board acc. | Total board acc. |
| RRN | 0.0 | 0.0 | 0.0 | 0.0 | 0.0 | 0.0 |
| SATNet | 0.0 | 0.0 | 0.0 | 0.0 | 0.0 | 0.0 |
| SATNet* | 72.7 | 75.9 | 67.3 | 75.7 | 0.1 | 0.1 |
| L1R32H4 | 94.1 | 91.0 | 90.5 | 87.7 | 65.8 | 65.7 |
| NTR | 87.4 | 0.0 | 0.0 | 91.4 | 3.9 | 3.9 |
| NDC | 79.9 | 0.0 | 0.0 | 88.0 | 0.0 | 0.01 |
| Ours | **95.5** | **95.9** | **95.5** | **93.1** | **94.4** | **93.1** |
| | Perception cell acc. | Solving cell acc. | Total cell acc. | Perception cell acc. | Solving cell acc. | Total cell acc. |
| RRN | 0.0 | 0.0 | 0.0 | 0.0 | 0.0 | 0.0 |
| SATNet | 0.0 | 0.0 | 0.0 | 0.0 | 0.0 | 0.0 |
| SATNet* | 99.1 | 98.6 | 98.8 | 75.7 | 59.7 | 72.0 |
| L1R32H4 | 99.8 | 99.1 | 99.4 | 99.3 | 89.5 | 92.6 |
| NTR | 99.7 | 60.1 | 77.8 | 99.7 | 38.5 | 57.3 |
| NDC | 99.4 | 10.8 | 50.4 | 99.5 | 10.9 | 38.7 |
| Ours | **99.9** | **99.6** | **99.7** | **99.7** | **98.3** | **98.7** |

the logical reasoning computes the final path from the symbol $\mathbf{z}$ and tags it on the second map as the output $\mathbf{y}$.

As a detailed reference, we select some results of path planning generated by different methods and plot them in Figure 7. We find that some correct properties are learned by our method. For example, given the point $\mathbf{y}_{34}$ in the path, we have the following connectivity:

$$(\mathbf{y}_{34} = s) + (\mathbf{y}_{34} = e) + \text{Adj}(\mathbf{y}_{34}) = 2,$$

which means that the path point $\mathbf{y}_{34}$ should be connected by its adjacent points. In addition, some distinct constraints are also learned, for example,

$$\mathbf{y}_{32} + \mathbf{z}_{32} + \mathbf{z}_{11} + \mathbf{z}_{01} = 1.$$

In this constraint, $\mathbf{z}_{11}$ and $\mathbf{z}_{01}$ are two noise points, and they always take the value of $0$. Therefore, it actually ensures that if $\mathbf{z}_{32}$ is an obstacle, then $\mathbf{y}_{32}$ should not be selected as a path point. However, it is still unknown whether our neuro-symbolic framework derives all the results as expected, because some of the learned constraints are too complex to be understood.

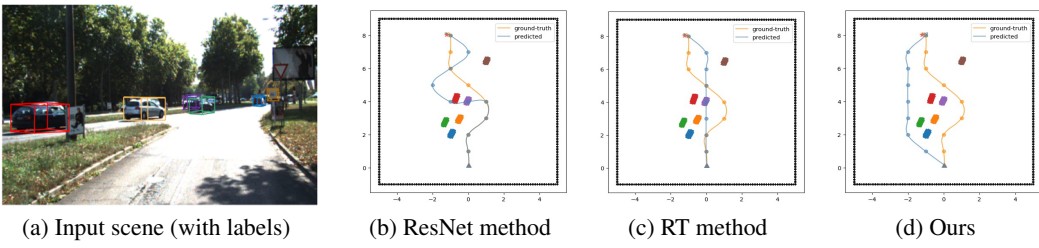

(a) Input scene (with labels) (b) ResNet method (c) RT method (d) Ours

Figure 7: Some results of neuro-symbolic learning methods in self-driving path planning task.

Table 6: Detailed cell and board accuracy (%) of **transfer** visual Sudoku task.

| Method | SATNet → RRN | | | RRN → SATNet | | |
|---|---|---|---|---|---|---|
| | Perception board acc. | Solving board acc. | Total board acc. | Perception board acc. | Solving board acc. | Total board acc. |
| RRN | 0.0 | 0.0 | 0.0 | 0.0 | 0.0 | 0.0 |
| SATNet | 0.0 | 0.0 | 0.0 | 0.0 | 0.0 | 0.0 |
| SATNet* | 80.8 | 1.4 | 1.4 | 0.0 | 0.0 | 0.0 |
| L1R32H4 | 84.8 | 21.3 | 21.3 | 94.9 | 95.0 | 94.5 |
| NTR | 90.2 | 0.0 | 0.0 | 86.9 | 0.0 | 0.0 |
| NDC | 86.1 | 0.0 | 0.0 | 82.4 | 0.0 | 0.0 |
| Ours | **93.9** | **95.2** | **93.9** | **95.2** | **95.3** | **95.2** |
| | Perception cell acc. | Solving cell acc. | Total cell acc. | Perception cell acc. | Solving cell acc. | Total cell acc. |
| RRN | 0.0 | 0.0 | 0.0 | 0.0 | 0.0 | 0.0 |
| SATNet | 0.0 | 0.0 | 0.0 | 0.0 | 0.0 | 0.0 |
| SATNet* | 99.1 | 66.2 | 76.5 | 65.8 | 53.8 | 59.2 |
| L1R32H4 | 99.3 | 89.5 | 92.6 | 99.7 | 99.6 | 99.7 |
| NTR | 99.6 | 37.1 | 56.3 | 99.6 | 62.4 | 79.0 |
| NDC | 99.4 | 11.0 | 38.7 | 99.5 | 11.3 | 50.7 |
| Ours | **99.8** | **98.4** | **98.8** | **99.8** | **99.7** | 99.7 |

