Table 3: Detailed cell and board accuracy (%) of **original** visual Sudoku task.

| Method | SATNet dataset | | | RRN dataset | | |
|---|---|---|---|---|---|---|
| | Perception board acc. | Solving board acc. | Total board acc. | Perception board acc. | Solving board acc. | Total board acc. |
| RRN | 0.0 | 0.0 | 0.0 | 0.0 | 0.0 | 0.0 |
| SATNet | 0.0 | 0.0 | 0.0 | 0.0 | 0.0 | 0.0 |
| SATNet* | 72.7 | 75.9 | 67.3 | 75.7 | 0.1 | 0.1 |
| L1R32H4 | 94.1 | 91.0 | 90.5 | 87.7 | 65.8 | 65.7 |
| NTR | 87.4 | 0.0 | 0.0 | 91.4 | 3.9 | 3.9 |
| NDC | 79.9 | 0.0 | 0.0 | 88.0 | 0.0 | 0.01 |
| Ours | **95.5** | **95.9** | **95.5** | **93.1** | **94.4** | **93.1** |
| | Perception cell acc. | Solving cell acc. | Total cell acc. | Perception cell acc. | Solving cell acc. | Total cell acc. |
| RRN | 0.0 | 0.0 | 0.0 | 0.0 | 0.0 | 0.0 |
| SATNet | 0.0 | 0.0 | 0.0 | 0.0 | 0.0 | 0.0 |
| SATNet* | 99.1 | 98.6 | 98.8 | 75.7 | 59.7 | 72.0 |
| L1R32H4 | 99.8 | 99.1 | 99.4 | 99.3 | 89.5 | 92.6 |
| NTR | 99.7 | 60.1 | 77.8 | 99.7 | 38.5 | 57.3 |
| NDC | 99.4 | 10.8 | 50.4 | 99.5 | 10.9 | 38.7 |
| Ours | **99.9** | **99.6** | **99.7** | **99.7** | **98.3** | **98.7** |

## G.4 Self-driving Path Planning

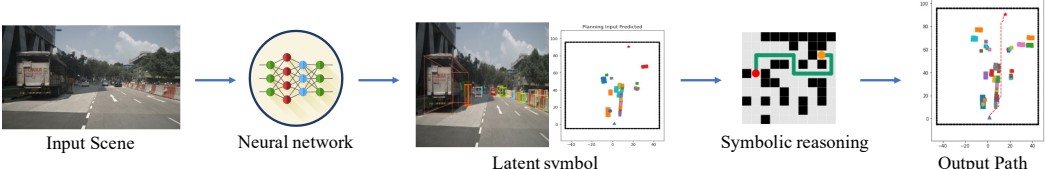

Figure 4: A neuro-symbolic system in self-driving tasks. The neural perception detects the obstacles from the image collected by the camera; the symbolic reasoning plans the driving path based on the obstacle map. The neuro-symbolic learning task is to build these two modules in an end-to-end way.

The goal of the self-driving path planning task is to train the neural network for object detection and to learn the logical constraints for path planning in an end-to-end way. As shown in Figure 6, we construct two maps and each contains $10 \times 10$ grids (binary variables). The neural perception detects the obstacles from the image $\mathbf{x}$ and locates it in the first map, which is essentially the symbol $\mathbf{z}$. Next, the logical reasoning computes the final path from the symbol $\mathbf{z}$ and tags it on the second map as the output $\mathbf{y}$.

As a detailed reference, we select some results of path planning generated by different methods and plot them in Figure 7. We find that some correct properties are learned by our method. For example, given the point $\mathbf{y}_{34}$ in the path, we have the following connectivity:

$$(\mathbf{y}_{34} = s) + (\mathbf{y}_{34} = e) + \mathrm{Adj}(\mathbf{y}_{34}) = 2,$$

which means that the path point $\mathbf{y}_{34}$ should be connected by its adjacent points. In addition, some distinct constraints are also learned, for example,

$$\mathbf{y}_{32} + \mathbf{z}_{32} + \mathbf{z}_{11} + \mathbf{z}_{01} = 1.$$

Table 4: Detailed cell and board accuracy (%) of **transfer** visual Sudoku task.

| Method | SATNet → RRN | | | RRN → SATNet | | |
| --- | --- | --- | --- | --- | --- | --- |
| | Perception board acc. | Solving board acc. | Total board acc. | Perception board acc. | Solving board acc. | Total board acc. |
| RRN | 0.0 | 0.0 | 0.0 | 0.0 | 0.0 | 0.0 |
| SATNet | 0.0 | 0.0 | 0.0 | 0.0 | 0.0 | 0.0 |
| SATNet* | 80.8 | 1.4 | 1.4 | 0.0 | 0.0 | 0.0 |
| L1R32H4 | 84.8 | 21.3 | 21.3 | 94.9 | 95.0 | 94.5 |
| NTR | 90.2 | 0.0 | 0.0 | 86.9 | 0.0 | 0.0 |
| NDC | 86.1 | 0.0 | 0.0 | 82.4 | 0.0 | 0.0 |
| Ours | **93.9** | **95.2** | **93.9** | **95.2** | **95.3** | **95.2** |
| | Perception cell acc. | Solving cell acc. | Total cell acc. | Perception cell acc. | Solving cell acc. | Total cell acc. |
| RRN | 0.0 | 0.0 | 0.0 | 0.0 | 0.0 | 0.0 |
| SATNet | 0.0 | 0.0 | 0.0 | 0.0 | 0.0 | 0.0 |
| SATNet* | 99.1 | 66.2 | 76.5 | 65.8 | 53.8 | 59.2 |
| L1R32H4 | 99.3 | 89.5 | 92.6 | 99.7 | 99.6 | 99.7 |
| NTR | 99.6 | 37.1 | 56.3 | 99.6 | 62.4 | 79.0 |
| NDC | 99.4 | 11.0 | 38.7 | 99.5 | 11.3 | 50.7 |
| Ours | **99.8** | **98.4** | **98.8** | **99.8** | **99.7** | 99.7 |

In this constraint, $z_{11}$ and $z_{01}$ are two noise points, and they always take the value of $0$. Therefore, it actually ensures that if $z_{32}$ is an obstacle, then $y_{32}$ should not be selected as a path point. However, it is still unknown whether our neuro-symbolic framework derives all the results as expected, because some of the learned constraints are too complex to be understood.

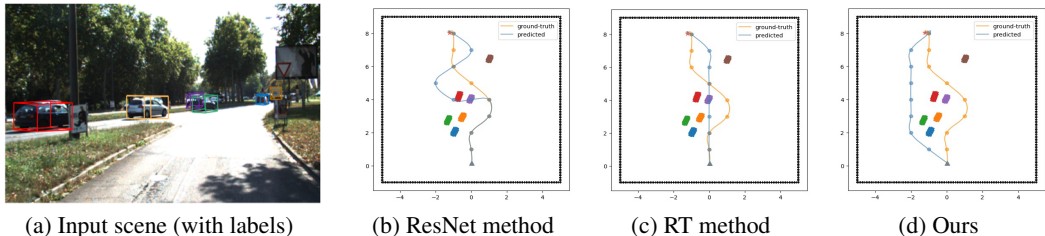

(a) Input scene (with labels)    (b) ResNet method    (c) RT method    (d) Ours

Figure 5: Some results of neuro-symbolic learning methods in self-driving path planning task.