# OpenReview forum: "Neuro-symbolic Learning Yielding Logical Constraints"
_NeurIPS.cc/2023/Conference — NeurIPS 2023 poster_

### Official Review · Reviewer_MqQr · 2023-06-29

**Soundness:** 3 good
**Presentation:** 2 fair
**Contribution:** 4 excellent
**Rating:** 7
**Confidence:** 3

**Summary:**

This paper takes inspiration from convex and bilevel optimization to derive a learning algorithm that simultaneously learns neural network perception and rules on the perceptions. The authors derive an efficient training algorithm involving multiple minimization steps.

Post rebuttal: I thank the authors for addressing the concerns. I will keep my score, this is a good paper, and I am happy error bars are included in the rebuttal.

**Strengths:**

The paper tackles the genuinely very hard problem of learning rules and perception simultaneously. The performance on, from what I can tell, quite challenging tasks is great. Ablations show multiple complex techniques help with performance.
Furthermore, the paper has several proofs on convergence and optimality.

The method itself has many (original) interesting ideas, although there are many.

**Weaknesses:**

While the paper is decently written, there are a lot of technicalities to the method that makes understanding the high-level picture challenging. It relies on many optimization methods: Cardinality constraints, difference of convex programming, proximal point algorithms and trust regions, not to mention quite a few hyperparameters. Some equations fly in without a clear justification, and glancing at the appendix does not help me either. Furthermore, the paper introduces a hyperparameter alpha that significantly complicates the computation without actually using it.

The main goal of this paper is to yield logical constraints, but whether they are also interpretable is not evaluated.

The paper does not have error bars on experiments.

**Questions:**

- An SMT solver is mentioned multiple times of which in Figure 1. Is my understanding correct that it is only used during inference, not training?
- Would the method also work by taking the cross entropy for the $\ell_1$ loss?
- Page 4, 131: Why is it allowed to rewrite the objective introducing alpha like that?
- Proposition 1: Explain what $\mathbf{e}$ is.
- Page 5, 154: The linearization uses $(e-2u)^Tu$, but further down (line 172) it is $(e-w)^Tw$. Where did the 2 go to?
- Same question for neural network training. Also, I think there's something wrong with the braces there.
- To what values do you set $m$ (number of rules)?
- Appendix, table 1: You mention that $\alpha$ is fixed to 1, meaning the influence from the rules is ignored by symbol grounding. Why did you choose this setting? $\alpha$ significantly complicates your mathematics. Setting it to 1 would simplify the equations significantly.

**Limitations:**

Inference of this method is very expensive, as highlighted in the limitations section. I'd have preferred this in the main paper, not in the appendix.

---

> ### Author Rebuttal · Authors · 2023-08-07
>
> ### **Response to Reviewer MqQr**
>
> Thanks for the comments.
>
> **The role of SMT solvers in our framework:** Yes, SMT solvers are only used during inference.
>
> **Replace $\ell_1$ loss by cross entropy:** Yes, we can directly change equation (6) to the gradient descent of cross entropy loss, and it should also work. This is not only implied by theory, but also observed by our preliminary experiments.
>
> **The derivation of the equation in Line 131:** The original constraint in network training is $\bar{\mathbf{z}} = \arg\min\nolimits_{\mathbf{z} \in \mathcal{Z}}~ \ell_2(h_{\boldsymbol{\boldsymbol{\phi}}}(\mathbf{z},\mathbf{y}), 1) := \\|\boldsymbol{W} (\mathbf{z}; \mathbf{y}) - \boldsymbol{b}\\|^2$.
> However, this constraint is intractable due to the ill-conditioned $\boldsymbol{W}$. Hence, we introduce the network prediction as a guidance, forming the final equation in Line 131.
>
> **Explain $\boldsymbol{e}$:**  $\boldsymbol{e}$ is an all-one vector, and we will clarify this in the revision.
>
> **The derivation in Line 154 and neural network training equation:** Thanks for pointing them out. They are typos (we also confirm that our submitted code is correct). We will carefully checked the paper again and correct them in the revision.
>
> **How to set $m$ (number of rules):** In our experiment, we directly set a large enough $m$ to  learn all possible logical constraints. For example, we set $m = 2000$ in the visual SudoKu solving task, where only 324 constraints are required. Our method returns 324 unique constraints and the rest are redundant ones. We will include it in the revision.
>
> **Hyperparameter $\alpha$:** Sorry for the confusion due to the inconsistency of $\alpha$ in the main text and the Appendix. In the main text, we use the following formulation for symbol grounding problem:
> \begin{equation}
> \bar{\mathbf{z}} = \arg\min\nolimits_{\mathbf{z} \in \mathcal{Z}}~ (1-\alpha) \\|\boldsymbol{W} (\mathbf{z}; \mathbf{y}) - \boldsymbol{b}\\|^2 + \alpha \\|\mathbf{z} - f_{\boldsymbol{\theta}}(\mathbf{x})\\|^2.
> \end{equation}
> In appendix, we instead use
> \begin{equation}
> \bar{\mathbf{z}} = \arg\min\nolimits_{\mathbf{z} \in \mathcal{Z}}~ \\|\boldsymbol{W} (\mathbf{z}; \mathbf{y}) - \boldsymbol{b}\\|^2 + \alpha \\|\mathbf{z} - f_{\boldsymbol{\theta}}(\mathbf{x})\\|^2.
> \end{equation}
> $\alpha = 1$ is set for the latter, which is equivalent to $\alpha = 0.5$ for the former (where the network prediction and constraints satisfaction are considered equally). We will correct them in the revision.
>
> **The interpretable logical constraints:** For visual SudoKu solving task, a logical constraint is in the form of "x_111 + x_121 + x_131 + x_141 + x_151 + x_161 + x_171 + x_181 + x_191 == 1", meaning that there should be a "1" in the first row (where x_123 means the entry of 1st-row, 2st-column is 3). We will add some learned logical constraints for both tasks in Appendix G4.
>
> **Error bars on experiments:** We briefly provide some results of total board accuracy as reference, and will include the full version in the revision.
>
> |         |    SATNet dataset |       RRN dataset |  SATNet $\to$ RRN |  RRN $\to$ SATNet |
> | ------- | ----------------: | ----------------: | ----------------: | ----------------: |
> | SATNet* | 67.3 ($\pm$ 2.36) |  0.1 ($\pm$ 0.10) |  1.4 ($\pm$ 0.14) |  0.0  ($\pm$ 0.0) |
> | L1R32H4 | 90.5 ($\pm$ 2.85) | 65.7 ($\pm$ 6.97) | 21.3 ($\pm$ 3.48) | 94.5 ($\pm$ 1.65) |
> | Ours    | 95.6 ($\pm$ 0.31) | 92.8 ($\pm$ 0.73) | 93.9 ($\pm$ 0.40) | 95.2 ($\pm$ 0.91) |
>
> **Limitation section:** In the revision, we will summarize the limatations into the main body of the paper as suggested.

---

> > ### Comment · Reviewer_MqQr · 2023-08-14
> >
> > Thanks for addressing the concerns. I will keep my score, this is a good paper and I am happy error bars are included in the rebuttal.

---

### Official Review · Reviewer_MGh8 · 2023-07-04

**Soundness:** 3 good
**Presentation:** 3 good
**Contribution:** 3 good
**Rating:** 7
**Confidence:** 3

**Summary:**

This work provides a neural framework that combines network training, symbol grounding, and logical constraint synthesis.
It utilizes the cardinality constraints to express the logical constraint learning and a DC penalty for constraint relaxation.
The evaluation demonstrates that this method outperforms state-of-the-art models including both SATNET and L1R32H4 by a large margin.


**Strengths:**

Originality: 4/5
Pros: This methodology introduces two novel important components, DC relaxation loss, and cardinality constraints, into the numerical learning framework.
Cons: One interesting component in this paper, mapping the learned numerical rules into the symbolic space is not quite clearly explained. By studying the related work section, I think the "Softened Symbol Grounding for Neuro-symbolic Systems"[1] seems quite related to the rule extraction part. I feel this work is probably not emphasized enough in the paper's related work section.

Quality: 4/5
Pro: The theory and properties come with all the proofs explained in the appendix, which is quite convincing. This work also reaches better than the SOTA performance for the two experiments.
Cons: The SATNET* uses Distilled LeNet as its underlying perceptual model, while you have used a recurrent transformer model. This seems an unfair comparison.


Clarity: 3/5
Pros: The math component is nicely defined and with well-explained definitions.
Cons:   It is hard to understand the process of extracting a constraint from its numerical form ($\mathbf{w}$ and $\mathbf{b}$).

Significance: 4/5
Pros: This work improves from the existing work by learning constraints along with ground truth perception, which is an important task in neural symbolic learning.
Cons: The learned constraints are in boolean form and thus not high level enough to generalize across different task variants.

[1] Li, Zenan, et al. "Softened Symbol Grounding for Neuro-symbolic Systems." The Eleventh International Conference on Learning Representations. 2022.

**Weaknesses:**

See Strength.

Minor comments:
1. Line 91: There are two papers coming from different groups, whose first author's names are both Li. It seems that the two works come from the same person in the article.
2. I would recommend putting the limitation section into the main paper.

**Questions:**

1. What is the limitation of using cardinality constraint? Is there a fundamental limitation other than the tricky usage where introducing auxiliary variables is required?
2. How are the 324 cardinality constraints calculated?
3. Suppose the number of the constraint is unknown for a new task, what should a user do? How will that impact the performance?
4. How to interpret a boolean-based constraint to an average programmer?

I am happy to raise my score if these questions are addressed.

**Limitations:**

Suggestions included in the Question section.

---

> ### Author Rebuttal · Authors · 2023-08-07
>
> ### **Response to Reviewer MGh8**
>
> Thanks for the comments. Following are our responeses.
>
> **Limitation of cardinality constraint:** Essentially, cardinality constraints can represent any propositional logic formula, i.e., they have the same expressiveness with CNF (Conjunctive Normal Form) or DNF (Disjunctive Normal Form) of propositional logic. However,  when learning cardinality constraint, some biases do exist (e.g., bounding box $[b_{min}, b_{max}]$ should be tight) which may result in the incompleteness of the learned constraints.
>
> **The calculation of 324 cardinality constraints:** A logical constraint "x_111 + x_121 + x_131 + x_141 + x_151 + x_161 + x_171 + x_181 + x_191 == 1" means that there should be a "1" in the first row (where x_123 means the entry of 1st-row, 2st-column is 3). We have 9 such constraints for the first row, and we have 9 rows, resulting a total of $9\times 9=81$ constraints for rows. We also have 81 constraitns for columns and blocks, respectively. There are also $9 \times 9=81$ constraints requiring that each cell should be in $1-9$. Therefore, the total number of constraints is $81 \times 4=324$. For our method, we initially set a large number, and then remove the redundant constraints, resulting exactly 324 contraints confirmed by manual checking.
>
> **How to determine the number of constraints:** One can directly set a sufficiently large $m$ (number of constraints), and we observe that a large $m$  does not ruin the effectiveness of our method. For example, we set $m = 2000$ in the visual SudoKu solving task, where only 324 constraints are required. Our method returns 324 unique constraints and the rest are redundant. A more efficient way is initially setting a large $m$ to estimate the actual number of logical constraints needed, and then adjusting it for more efficient training. We will include a brief discussion on the setting of $m$ in the revision (Appendix F).
>
> **Interpret a boolean-based constraint to an average programmer:** A reasonable method is to translate the Boolean-based constraints into a CNF formula, and we think it is more friendly for an average programmer to understand such a CNF formula. The tranlsation is studied for a long time and can be automatically conducted by off-the-shelf SMT solvers [1, 2].
>
> **Relation to the paper “Softened symbol grounding…”:** We will highlight the relation in the revision as suggested.
>
> **Compare with SATNet using LeNet architecture:** We agree that it could be unfair to compare with SATNet* considering different architectures. As an ablation study, we use LeNet-5 model in our method, deriving the results of total board accuracy as follows.
>
> |                | SATNet dataset | RRN dataset |    SATNet $\to$ RRN |     RRN $\to$ SATNet |
> | ---------------- | ---------------: | ----------: | ---------------: | ---------------: |
> | SATNet*        |           67.3 |         0.1 |              1.4 |              0.0 |
> | Ours (LeNet-5) |           75.2 |        79.6 |             82.4 |             70.6 |
>
> We also observe that our method successfully learned full logical constraints, but the accuracy of perception module degrades from 99.8% to about 99.1%, leading to the drop of overall accuracy.
>
> **Misleading citation in Line 91:** We will correct it in the revision.
>
> **Limitation section:** In the revision, we will summarize the limatations into the main body of the paper.
>
> [1] Sinz, C. (2005, October). Towards an optimal CNF encoding of boolean cardinality constraints. In International conference on principles and practice of constraint programming (pp. 827-831). Berlin, Heidelberg: Springer Berlin Heidelberg.
>
> [2] Eén, N., & Sörensson, N. (2006). Translating pseudo-boolean constraints into SAT. Journal on Satisfiability, Boolean Modeling and Computation, 2(1-4), 1-26.

---

> > ### Comment · Reviewer_MGh8 · 2023-08-13
> >
> > Thank you for addressing my concerns.

---

### Official Review · Reviewer_w2hx · 2023-07-30

**Soundness:** 3 good
**Presentation:** 3 good
**Contribution:** 3 good
**Rating:** 7
**Confidence:** 1

**Summary:**

The authors propose a fusion of neural network and symbolic domain via logical constraints to learn specific vision tasks in a weakly supervised way. They break it down to two optimization problems for neural and symbolic domains. The logical constraints are solved using  a deterministic solver and grounded softly to the neural component via a latent space $z$.

**Strengths:**

* The neuro-symbolic approach of solving tasks that require some planning is always an interesting direction.
* The results on the two domains studied in the paper are quite strong, indicating that such fusion can indeed converge well

**Weaknesses:**

* The paper needs more high level figures to help understand the exact logical constraints being enforced for both the domains. This will significantly improve the readability.
* Please add some discussion about the limitations of using the specific cardinality-based logical constraints you chose in terms of where it might be applicable vs where it cannot.

**Questions:**

* Why do you chose these specific tasks? Is it following some prior works? Please make it clear. This will help understand the scope of the work in terms of applicability to different domains.

**Limitations:**

* The authors don't have any limitations section. Please refer to weakness for recommendations on this.

---

> ### Author Rebuttal · Authors · 2023-08-07
>
> ### **Response to Reviewer w2hx**
>
> Thanks for the comments.
>
> **The motivation of tasks:** The visual SudoKu solving task is a standard and commonly-evaluated task from existing neuro-symbolic learning methods. The self-driving planning task is proposed by ourselves. Specifically, we aim to introduce this more practical task for the neuro-symbolic learning community. We will further clarify this in Appendix G3 and G4.
>
> **Limitation:** We discussed the limitations in Appendix A. In the revision, we will summarize the limitations into the main body of the paper.

---

> > ### Comment · Reviewer_w2hx · 2023-08-13
> > **Response to authors**
> >
> > Thanks for clarifying my concerns.

---

### Official Review · Reviewer_X5tc · 2023-07-30

**Soundness:** 3 good
**Presentation:** 2 fair
**Contribution:** 3 good
**Rating:** 5
**Confidence:** 3

**Summary:**

The paper proposes a new neurosymbolic approach for learning symbolic representations and logical constraints on top of these simultaneously. The authors propose a new penalty term based on Difference of Convex (DC) programming in order to relax the optimzation problem. The performance of the proposed approach is demonstrated on the visual sudoku challenge and a path planning environment based on the Kitti and NuScene datasets. Overall, the method performs better than existing works and achieves results close to a fully supervised architecture in some cases.

**Strengths:**

The paper takes a principled approach to the problem of learning symbolic representations and logic rules simultaneously. The proposed DC programming approach allows for the theoretical analysis as per Theorem 1 demonstrating that gradual increase of the weight of the DC penalty term results in the desired stationary point.
Also, the reported results demonstrate strong improvement over other neurosymbolic approaches in the two considered environments of the visual sudoku and path planning.

**Weaknesses:**

The performed experiments focus on 2 relatively simple and static environments. Moreover, one of the core motivations behind neurosymbolic approaches is the ability to learn faster and/or transfer knowledge to new environments, however none of these questions have been addressed by the conducted experiments. Overall, I think the paper would benefit from making the methodology section more concise and to the point and expanding more on the results.

**Questions:**

- What does the notation (z;y) mean exactly in w^T(z;y)? Is it some form of concatenation?
- You mention that the solvers can find a solution even with incorrect perception. Why do you think this happens? Do you think that using more powerful solvers can actually be a problem for the perception side of things?

**Limitations:**

The paper does not discuss the limitations of the proposed methodology which is important in order to identify meaningful directions for future work.

---

> ### Author Rebuttal · Authors · 2023-08-07
>
> ### **Response to Reviewer X5tc**
>
> Thanks for the comments.
>
> **The challenge of two tasks:** The visual SudoKu solving task is a standard and commonly-evaluated task in existing neuro-symbolic learning methods. Particularly, only 17 out of 81 cells are initially filled in some of the SudoKu boards of the RRN dataset, and the difficulty of solving such a SudoKu can be well understood in the Appendix of  [1]. Furthermore, we try to explore some real applications of  neuro-symbolic learning via introducing the self-driving planning task. In this task, we extract two main modules (i.e., object localization and path planning) in self-driving systems, which is suitable for evaluating neuro-symbolic learning method (object localization is the neural part and path planning is the symbolic part).
>
> We would also like to point out that, to the best of our knowledge, the genuine end-to-end learning of  neuro-symbolic systems has not been achieved before, even for simple (exemplar) tasks such as those used in the evaluation. This paper shows that it can be made feasible with a natural and disciplined approach, including a game theory framework connecting neural and symbolic learning, as well as more techical approaches such as trust-region penalty (to prevent degeneracy), and DC relaxation (to preserve logic exactness).
>
> **Knowledge transfer to new environments:** We agree with the reviewer on the core  motivations behind neuro-symbolic approaches. Actually, for knowledge transfer, we have conducted an experiment on transferring the learned rules from the SATNet dataset to the RRN dataset, and vice versa. Note that these two datasets vary significantly in terms the difficulty levels of Sudoku solving tasks (RRN dataset is more challenging; see Section 4.1). Our experimental results (the third-column in Table 1 and Table 6 in Appendix G.3) show that, when applying both model and learned rules from SATNet (RRN) data to RRN (SATNet) data, our method is consistently effective, while the performance of existing methods drops dramatically.
>
> **Notation $(\mathbf{z};\mathbf{y})$:** We represent the column concatenation of two vectors $\mathbf{z} \in \mathbb{R}^m$ and $\mathbf{y} \in \mathbb{R}^n$ by $(\mathbf{z}; \mathbf{y}) \in \mathbb{R}^{m+n}$. We will clarify it in the revision.
>
> **The MAXSAT solver can find a correct solution despite incorrect perception:** The MAXSAT solver returns the solution achieving the highest number of satisfied constraints. Therefore, even with perception errors causing the full set of  logical constraints unsatisfiable, MAXSAT solvers may still output the correct results by satisfying a subset of the logical constraints.
>
> **Limitation:** We discussed the limitations in Appendix A. In the revision, we will summarize the limatations into the main body of the paper.
>
> [1] Palm, R., Paquet, U., & Winther, O. (2018). Recurrent relational networks. Advances in neural information processing systems, 31.

---

### Author Rebuttal · Authors · 2023-08-08

### **General response to reviewers**

We would like to thank all the reviewers for their kind and helpful feedback.

We start by clarifying that we have indeed discussed the limitations of our approach which is included in Appendix A. We will summarize and include them in the main paper.

We thank the reviewers for the suggestions on improving readability. We will give more intuitive explanations of the core ideas and the results in the revised paper, and move some technical details to the Appendix.

---

### Decision · Program_Chairs · 2023-09-21

**Decision:**

Accept (poster)

**Comment:**

This paper tackles the important problem of learning logical constraints in neurosymbolic AI. The optimization method proposed appears to work well on standard benchmarks.